# TOPA: Extending Large Language Models for Video Understanding via Text-Only Pre-Alignment

**Wei Li**[1,2]  **Hehe Fan**[1]*  **Yongkang Wong**[3]  **Mohan Kankanhalli**[3]  **Yi Yang**[1,2]

[1] ReLER Lab, CCAI, Zhejiang University, China
[2] The State Key Laboratory of Brain-Machine Intelligence, Zhejiang University, China
[3] School of Computing, National University of Singapore, Singapore
{weili6,hehefan,yangyics}@zju.edu.cn
yongkang.wong@nus.edu.sg    mohan@comp.nus.edu.sg
https://github.com/dhg-wei/TOPA

## Abstract

Recent advancements in image understanding have benefited from the extensive use of web image-text pairs. However, video understanding remains a challenge despite the availability of substantial web video-text data. This difficulty primarily arises from the inherent complexity of videos and the inefficient language supervision in recent web-collected video-text datasets. In this paper, we introduce Text-Only Pre-Alignment (TOPA), a novel approach to extend large language models (LLMs) for video understanding, without the need for pre-training on real video data. Specifically, we first employ an advanced LLM to automatically generate *Textual Videos* comprising continuous textual frames, along with corresponding annotations to simulate real video-text pairs. Then, these annotated textual videos are used to pre-align language-only LLMs with the video modality. To bridge the gap between textual and real videos, we employ the CLIP model as the feature extractor to align image and text modalities. During text-only pre-alignment, the continuous textual frames, encoded as a sequence of CLIP text features, are analogous to continuous CLIP image features, thus aligning the LLM with real video representation. Extensive experiments, including zero-shot evaluation and finetuning on various video understanding tasks, demonstrate that TOPA is an effective and efficient framework for aligning video modality with LLMs. In particular, without training on any video data, the TOPA-Llama2-13B model achieves a Top-1 accuracy of 51.0% on the challenging long-form video understanding benchmark, EgoSchema. This performance surpasses previous video-text pre-training approaches and is competitive with recent GPT-3.5-based video agents.

## 1   Introduction

Image-language understanding has made large advancements in both image-language alignment [26, 48] and Multimodal Large Language Models (MLLMs) [1, 27, 34, 89], aided by pre-training on large-scale noise-paired image-text data collected from the web [6, 19, 51, 53, 52]. This raises a question: *Can we mirror this success in video-language understanding?* Research [45, 64, 77, 86] has explored pretraining video-language models on millions of web video-text data [3, 40, 65], achieving promising results in basic video tasks such as video-text retrieval, video captioning, and video question answering across conventional video benchmarks. However, recent research reveals that these models struggle with a challenging long-form video understanding benchmark, *i.e.*, EgoSchema [39], which requires intrinsic temporal understanding capabilities. This highlights the gap in adapting web video-text pretrained models to more comprehensive video understanding tasks.

---

*Hehe Fan is the corresponding author.

38th Conference on Neural Information Processing Systems (NeurIPS 2024).

We attribute this gap to two primary factors: *1) The intrinsic complexity of the video modality.* Videos introduce intrinsic complexities in both spatial and temporal dimensions, which are not present in static images. These complexities require extensive training on larger-scale data to effectively capture video dynamics. Furthermore, representing videos typically involves processing multiple frames, significantly increasing computational demands compared to image modeling. The dual challenges of large-scale training and increased computational requirements make video-language modeling particularly challenging. *2) The limitations of web language supervision.* The language supervision in recent web video-text datasets primarily comes from subtitles or descriptions associated with the videos [3, 40]. However, subtitles often suffer from the issues of visual-textual misalignment [33, 17]. Moreover, the form of descriptive supervision is inefficient in building robust video reasoning capabilities, especially in terms of temporal reasoning. This mismatch between the complex video content and the limited supervision hinders effective video-language modeling.

In this paper, we propose an innovative approach to develop video understanding capabilities by using LLMs to simulate and understand video dynamics. Instead of directly aligning LLMs with real video representation, we first introduce a textual video representation — a sequence of textual frames designed to mimic real visual dynamics. This textual video can be readily generated by advanced LLMs and effectively simulates various video dynamics by describing them in text. Specifically, we present a Textual Video (TextVid) dataset, automatically generated by LLMs. TextVid includes: 1) *Textual videos* (hereinafter referred to as "**Tideo**"), which consist of a sequence of textual frames crafted to mimic the keyframes of real videos, and 2) *Tideo annotations*, including comprehensive Tideo-level dense descriptions and varied question-answer (QA) pairs. These annotations are of high quality and closely align with the Tideo content, by virtue of the powerful capability of LLM in language generation.

Building on the proposed TextVid dataset, we introduce the Text-Only Pre-Alignment (TOPA) framework, to effectively and efficiently pre-align LLMs with the video modality, reducing the need for costly video-text pre-training. We introduce three tasks for video-LLM pre-alignment: Tideo summarization, Tideo QA and multi-choice Tideo QA. To bridge the gap between textual Tideos and visual videos, we leverage the CLIP [48] model for feature extraction. Specifically, we employ the CLIP text encoder to extract frame-level representations for Tideos, and the CLIP visual encoder for real videos. During the text-only pre-alignment phase, the LLM learns to process continuous CLIP text features of Tideos. In the real video inference phase, it transitions to handling continuous CLIP image features of real video. Due to the aligned CLIP image-text feature space, the LLM can adapt to real video inputs despite being trained on textual representations. Our main contributions include:

(1) We propose a novel Text-Only Pre-Alignment (TOPA) framework to extend Large Language Models (LLMs) for video understanding. TOPA aligns LLMs with the video modality efficiently and effectively without the need for training on real videos, reducing the costs for video-text pre-training.

(2) We introduce TextVid, a textual video dataset automatically generated by advanced LLMs. TextVid dataset comprises 721K diverse Tideos along with associated high-quality annotations, which include detailed Tideo descriptions and a variety of question-answer pairs.

(3) Extensive experiments demonstrate TOPA's effectiveness across various video understanding tasks. Particularly, the TOPA-Llama2-13B model achieves 51.0% Top-1 accuracy in the challenging EgoSchema benchmark, outperforming previous video-text pretraining methods and competitive with recent GPT-3.5-based video agents.

## 2 Related Work

**Vision-language alignment.** CLIP [48] aligns the vision and language modalities in a common feature space via contrastive learning with large-scale web image-text data. MLLMs [1, 27, 34, 89] align the visual model with LLM via training on image-caption pairs and interleaved image-text data. Video-LLMs [7, 23, 32, 83] explore modeling video sequences within LLM spaces, leveraging LLM for video-language understanding. In this paper, we focus on video-LLM alignment. Rather than using multimodal data for vision-language alignment, we introduce a novel text-only pre-alignment framework to extend LLMs for video understanding without pre-training on real video-text data.

**LLMs for multimodal data augmentation.** Recent research explores the use of LLMs to enhance the multimodal data. A line of work [5, 12, 34] use LLMs for refining captions or extending the

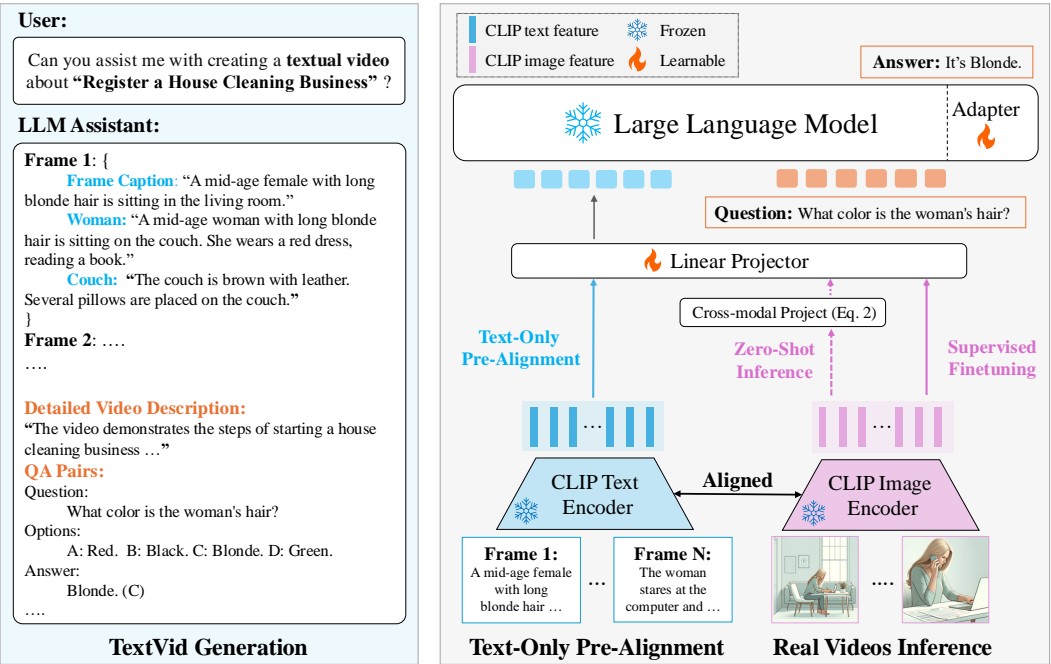

Figure 1: Overview of the proposed Text-Only Pre-Alignment (TOPA) framework. *Left*: The pipeline used for generating the TextVid dataset. *Right*: The video-LLM alignment framework. During text-only pre-alignment, the LLM learns to process continuous CLIP text features. In zero-shot inference, the LLM uses projected CLIP visual features as input. Additionally, TOPA supports supervised fine-tuning on downstream video datasets to further improve the performance.

image caption pairs to diverse visual tasks like visual conversation and image editing. Another line of work [28, 37, 38, 47] further employ advanced LLM to enrich web video supervision for video instruction tuning. In this paper, rather than enhancing multimodal datasets, we propose generating text-only data consisting of "textual videos" and diverse language supervision, which aims to simulate real videos and their corresponding annotations.

**Long-form video understanding.** Long-form video understanding [39, 61, 71] presents significant challenges due to the intricate spatial and temporal dynamics. Conventional video-text pretraining approaches [4, 45, 65, 66, 77, 90] utilize extensive web video-caption data for video-language alignment. Recent research [28, 66, 83, 88] employ video instruction-tuning for video-LLM alignment to enhance video-language understanding. Another line of research [22, 54, 81, 49] seeks to adapt recent image MLLMs to video understanding. A parallel line of research [9, 42, 55, 60, 63, 13, 78, 82, 67, 21] combine the LLM with various VLM tools as video agents to perform video-understanding tasks. In this paper, we propose a novel text-only pre-alignment framework to efficiently and effectively align LLMs with videos without pre-training on real videos.

## 3  Method

In this section, we detail the TOPA framework. We first introduce the data generation pipeline of TextVid Dataset (Section 3.1). Next, we describe how to align the Tideo representation with LLM (Section 3.2). Finally, we discuss adapting the text-only aligned video-LLM model for real video inference (Section 3.3). An overview is illustrated in Figure 1.

### 3.1  TextVid Dataset

This dataset, comprising textual videos (Tideos) and associated annotations, is generated by an advanced LLM (*i.e.*, Gemini Pro 1.0 [56]). The data generation pipeline is detailed in Appendix D. Each Tideo is presented in a textual format and contains 5-15 sequential frames. Each frame includes a frame caption that describes the scene and multiple object captions. To enhance understanding and

interaction with these Tideos, the dataset features a dense description summarizing the Tideo, as well as a set of multiple-choice questions and answers related to the Tideo content. The structure of each element is as follows:

> **Dataset Element**:
>  **Tideo**: Sequence of textual frames $\{T_1, T_2, \ldots, T_n\}, 5 \leq n \leq 15$
>    For each frame $T_i$:
>      Frame caption: $C_i$
>      Object captions: $D_{i,j}$ for main objects in $T_i$
>  **Annotations**:
>    Global Dense Description of the Tideo: $D_V$
>    Set of Questions-Options-Answers: $\{(Q_k, O_k, A_k)\}$

There are two major advantages of the TextVid dataset. **(1) The large-scale and diverse Tideos.** As the dataset is text-only and fully generated by an LLM, the size of TextVid is practically unlimited. Moreover, the Tideos can cover a broad range of domains by simply prompting the language model with appropriate conditions. It is distinctly different from previous web video-text dataset like Howto100M [40] that are limited to specific human-centric instructional videos. In practice, we enhance the diversity of TextVid by randomly sampling video captions from WebVid-2M [3], video titles from Howto100m [40], video tasks from Ego4D [15] and object names with descriptions from WordNet [41] as a condition of prompts. These varied prompts enable the language model to generate a diverse dataset. **(2) The high-quality, consistent and free-form language supervision.** The language supervisions are generated along with Tideos. The advanced capabilities of LLM ensure the quality of these supervisions, making them less noisy than web video-text data. Moreover, both the Tideo and the supervision are in textual format, making the supervision closely aligned with the Tideo's content. Additionally, the format of the language supervision is unrestricted. For example, we prompt the LLM to generate dense descriptions and multi-choice QA pairs as language supervision.

## 3.2 Text-Only Pre-Alignment

**Preliminary: Video-LLM alignment.** The goal of video-LLM alignment is to extend pre-trained LLMs for processing video inputs. Given a video sampled with $n$ frames $\{\mathbf{I}_1, \mathbf{I}_2, \ldots, \mathbf{I}_n\}$, Recent work [23, 77] uses a frozen CLIP model to extract the frame-level visual feature, formulated as $\mathbf{f}_i^v = E_{\text{image}}(\mathbf{I}_i)$, where $E_{\text{image}}$ denotes CLIP image encoder. The CLIP features are then projected into the LLM space via a simple linear layer, denoted as $G(P(\mathbf{f}_1^v), ..., P(\mathbf{f}_n^v))$, where $G$ denotes a language model and $P$ denotes a projection layer that projects the CLIP feature to LLM space.

**Tideo representation.** In this work, we leverage Tideos (*c.f.* Section 3.1) for video-LLM pre-alignment instead of training on real videos. Specifically, given the textual frame $T_i$, we employ CLIP text encoder to extract the frame representation from frame caption $C_i$ and detailed object captions $D_i$, represented as $\mathbf{f}_i^t = F_{\text{fusion}}(E_{\text{text}}(C_i), E_{\text{text}}(D_{i,1}), ..., E_{\text{text}}(D_{i,j}))$, where $F_{\text{fusion}}$ is a fusion function such as simple average pooling, and $E_{\text{text}}$ denotes the CLIP text encoder. A Tideo with $n$ textual frames is represented as $\mathbf{V}^t = \{\mathbf{f}_1^t, ..., \mathbf{f}_n^t\}$.

**Text-only pre-alignment.** Given the Tideo $T$, dense Tideo-level description $D_V$, and QA pairs with multiple choices $\{(Q_k, O_k, A_k)\}$, we introduce the following tasks for Tideo-LLM alignment: (1) **Tideo Summarization**: Given the Tideo, generate a detailed description to summarize the Tideo; (2) **Tideo QA**: Given the Tideo and question, predict the answer; (3) **Multi-choice Tideo QA**: Given the Tideo, question and multiple choices, choose the correct answer from the candidates. We employ a unified auto-regressive Language Modeling (LM) objective for these three tasks:

$$\mathcal{L}_{\text{LM}}(\theta_G, \theta_P) = -\frac{1}{|t|} \sum_{i=1}^{|t|} \log G(t_i | P(\mathbf{V}^t), Z, t_{<i}), \tag{1}$$

where $\mathbf{V}^t$ denotes the Tideo representation, and $\mathbf{V}^t = \{\mathbf{f}_1^t, ..., \mathbf{f}_n^t\}$ during the text-only training, Z denotes the task specific condition tokens and $t_i$ denotes the $i_{th}$ target token. $\theta_G$ and $\theta_P$ denote the learnable parameters of the LLM adapter and the projection layer $P$, respectively. In practice, we use the following format as the LLM input: {Task Instruction}. Video:$\{\mathbf{f}_1^t, ..., \mathbf{f}_n^t\}$. {Task Conditions}. Answer: {Predict Targets}. For the Tideo summarization task, the target is detailed Tideo descriptions. For Tideo QA task, the target is the answer and the condition is the question. For multi-choice Tideo

QA task, the target is the correct option and the condition consists with question and options. The details of the task-specific prompts are included in Appendix F.1.

### 3.3 Adapting to Real Video Understanding

Section 3.2 introduces the text-only pre-alignment using the TextVid dataset. In this section, we detail how to adapt this text-only pre-aligned LLM for real video understanding. We introduce two approaches: one is zero-shot inference, which directly infers with real video data. And the other is supervised finetuning, where the pre-aligned model is further finetuned on downstream video data.

**Zero-shot inference.** During pre-alignment, we leverage the textual representation $\mathbf{V}^t = \{\mathbf{f}_1^t, ..., \mathbf{f}_n^t\}$ as the Tideo representation. During inference, we take real videos features as input, *i.e.*, $\mathbf{V}^v = \{\mathbf{f}_1^v, ..., \mathbf{f}_n^v\}$, where $\mathbf{f}_i^v = E_{\text{image}}(\mathbf{I}_i)$. These two modality features $\mathbf{f}^t$ and $\mathbf{f}^v$ that come from CLIP image encoder and CLIP text encoder are aligned via CLIP pre-training. This aligned image-text representation makes it possible to perform zero-shot inference without additional finetuning. However, the *modality gap* phenomenon [16, 30, 31, 44, 85], *i.e.*, CLIP image feature and CLIP text feature are located in two completely separate regions of the feature space, prevents us from directly taking the visual feature $\mathbf{f}^v$ as the textual feature $\mathbf{f}^t$. To bridge this modality gap, we follow DeCap [30] to employ a support memory to project the CLIP visual feature into the CLIP text feature space. This training-free projection process is formulated as:

$$\mathbf{f}^{v \to t} = \sum_{i=1}^{N} w_i * \mathbf{m}_i = \sum_{i=1}^{N} \frac{\exp((\mathbf{m}_i^\top \mathbf{f}^v)/\tau)}{\sum_{k=1}^{N} \exp((\mathbf{m}_k^\top \mathbf{f}^v)/\tau)} * \mathbf{m}_i, \tag{2}$$

where $\mathbf{m}_i$ denotes CLIP text feature from a pre-constructed memory of size $N$, $\mathbf{f}^v$ denotes input frame feature of real video and $\mathbf{f}^{v \to t}$ denotes the projected feature. During zero-shot inference, we take the $\mathbf{V}^{v \to t} = \{\mathbf{f}_1^{v \to t}, ..., \mathbf{f}_n^{v \to t}\}$ as the real video's representation.

**Supervised finetuning.** On the other hand, the text-only pre-alignment can be viewed as a pretraining stage. Following the pretraining-finetuning paradigm, the pre-aligned LLMs can then be fine-tuned on real video data for improved downstream task performance. The finetuning process is similar to the text-only pre-alignment as detailed in Section 3.2, except that the LLM receives a sequence of CLIP visual features as input instead of CLIP textual features.

### 3.4 Implementation Details

We leverage Llama2-7B, Llama2-13B [57] and Llama3-8B as the LLM backbone. Additionally, we employ the Llama-adapter [84] with an adaptation embedding length of 50. We utilize CLIP-ViT-L as the multimodal encoder. We employ a simple linear layer to project the CLIP feature into the LLM feature space. During training, the CLIP model and LLM backbone are frozen. The projection layer and additional Llama-adapter are trainable. For text-only pre-alignment, we uniformly sample the Tideos into 10 frames. We train the model on a mixture of tasks comprising Tideo summarization, Tideo QA, multi-choice Tideo QA with the ratio of 1:1:2. For zero-shot inference, we construct a memory for cross-modal projection, consisting of 2M CLIP text features sampled from captions in the TextVid dataset. TOPA-Llama2-7B and TOPA-Llama3-8B are trained on four 40G-A100 GPUs in one day. TOPA-Llama2-13B is trained in two days. More training details of TOPA and baselines are included in Appendix E.2.

## 4 Experiments

TOPA enables the LLM to perform various video understanding tasks as shown in Figure 2. In this section, we evaluate TOPA on multi-choice video QA and video captioning tasks. Section 4.1 evaluates TOPA on NeXT-QA [72], STAR [70], TVQA [24], recent challenging EgoSchema [39] and MVBench[29] benchmarks with the zero-shot setting. We further evaluate TOPA on multi-choice video QA with the finetuning setting (Section 4.2) and zero-shot video captioning task (Section 4.3). In Section 4.4, we conduct ablation study on the LLM prior and input video frames. We report Top-1 accuracy on multi-choice video QA benchmarks and CIDEr [58] score on video captioning benchmarks. We mainly compare TOPA with the following categories of video understanding approaches:

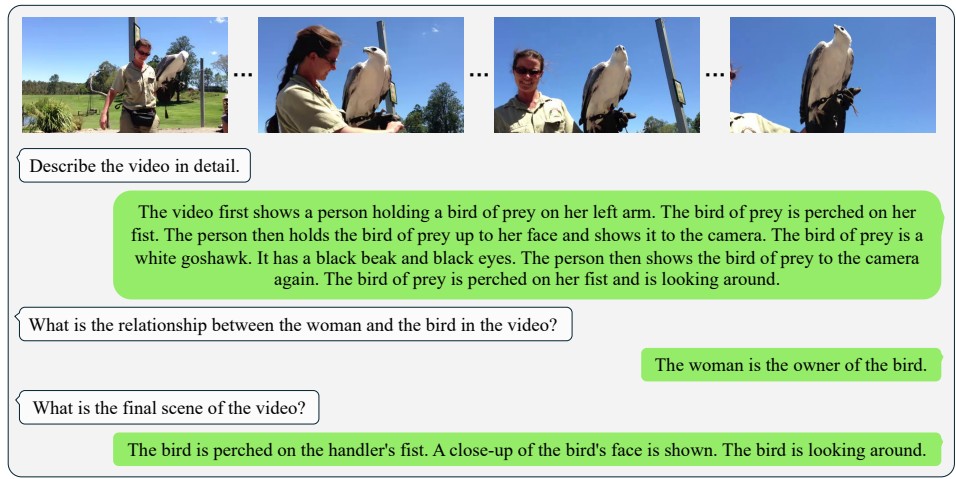

Figure 2: Examples of TOPA-LLama2-13B for video-language understanding. Given a video, TOPA is able to summarize the video content and answer the questions.

**(1) Web video pre-training approaches** [4, 45, 64, 66, 77]. This line of work aims to develop general video-language models by leveraging extensive web videos, using associated video captions or audio as weak supervision signals.

**(2) Adapting image MLLMs for video understanding** [22, 81, 49]. These approaches aim to extend the image understanding capabilities of recent vision-language models (VLMs) to video understanding. Specifically, SeViLa [81] utilizes BLIP-2 for localizing and understanding key frames of a video. IG-VLM [22] converts video into a composite image by arranging the video frames into a grid layout.

**(3) LLM-based video agents** [42, 59, 60, 63, 82, 13, 21, 67]. This line of work leverages LLMs like GPT-3.5 and GPT-4 as an agent to understand a video by designing and executing a series of actions. The language-only agents perceive visual information via recent foundation VLMs (*e.g.*, CLIP [48], BLIP-2 [27], LaViLa [87] and PALI [8]).

**(4) Our text-only pre-alignment.** Different from the above works, TOPA leverages the proposed TextVid dataset for video-LLM pre-alignment, enabling the LLM to process continuous features. Thus, it can enable performing video understanding tasks.

## 4.1 Zero-Shot Evaluation on Multi-Choice Video QA

### 4.1.1 Zero-shot Results on EgoSchema

Table 1 shows the results on EgoSchema full set. We compare our method against a range of recent approaches in video understanding. Our proposed text-only pre-alignment framework, despite training without real videos, shows impressive results on the EgoSchema benchmark. TOPA outperforms previous image-based adaptation approach IG-VLM and video agents LLoVi and Vamos with the same scale LLM (Llama2-7B and Llama2-13B). Moreover, TOPA shows consistent improvements when scaled up with a larger LLM backbone, indicating the effectiveness of LLMs in complex video understanding tasks.

**Discussion 1: The necessity of high-quality language supervision for video understanding.** Recent video pre-training approaches like LongViVit [45] and InternVideo [64], despite training on million-level web video-text data, show inferior performance on EgoSchema evaluation. These results highlight the inefficacy and inefficiency of conventional contrastive pre-training in understanding long-form videos, primarily due to noisy and simplistic language supervision. In contrast, our TOPA, trained on 721K Tideoswith high-quality language supervision, shows impressive results on EgoSchema. It indicates that, unlike image understanding which significantly benefits from leveraging web language as supervision, video understanding may require more precise and accurate language supervision to better capture the complex visual dynamics.

Table 1: Zero-shot results on EgoSchema [39] full set. Methods that leverage closed-source LLMs are marked in gray. † denotes the model is trained with in-domain egocentric videos from Ego4D [15]. * denotes results on EgoSchema subset. Results of InternVideo and FrozenBiLM are sourced from [39]. Results of SeViLA are sourced from [45].

| | Core VLMs | Core LLMs | Acc@1 |
|---|---|---|---|
| Human Eval [39][NeurIPS 2023] | - | - | 75.0 |
| Gemini-1.5-Pro [50][arXiv 2024.2 ] | - | Gemini-1.5-Pro | 63.2 |
| *(Pre-train on web video-text data)* | | | |
| FrozenBiLM [77][NeurIPS 2022] | - | - | 26.9 |
| InternVideo [64][arXiv 2022.12] | - | - | 32.1 |
| LongViViT [45][CVPR 2024 ] | - | - | 33.3 |
| MC-ViT-L† [4][ICML 2024 ] | - | - | 44.4 |
| InternVideo2$_{s3}$-6B† [66][arXiv 2024.3 ] | - | - | 41.1 |
| *(Adapt image MLLMs for video understanding)* | | | |
| SeViLA [81][NeurIPS 2023] | BLIP-2 | FLAN-T5-XL [10] | 22.7 |
| MVU [49][arXiv 2024.3 ] | LLaVA-v1.5-13B | Vicuna-13B | 37.6 |
| IG-VLM [22][arXiv 2024.3 ] | LLaVA v1.6-7B | Vicuna-7B | 35.8* |
| IG-VLM [22][arXiv 2024.3 ] | LLaVA v1.6-13B | Vicuna-13B | 47.0* |
| *(LLM-based video agents)* | | | |
| LangRepo [21][arXiv 2024.3 ] | CLIP-ViT-L | Mixtral-12B [20] | 41.2 |
| Vamos [60][arXiv 2023.11] | BLIP-2 | Llama2-13B | 36.7* |
| Vamos [60][arXiv 2023.11] | BLIP-2 | GPT-3.5 | 41.2* |
| Vamos [60][arXiv 2023.11] | BLIP-2 | GPT-4 | 48.3* |
| MoReVQA [42][CVPR 2024 ] | PALI-3-5B [8] | PaLM-2 [2] | 51.7 |
| LLoVi [82][arXiv 2024.3 ] | LaViLa† | GPT-3.5 | 50.3 |
| VideoAgent [63][ECCV 2024 ] | LaViLa† | GPT-4 | 54.1 |
| LifelongMemory [67][arXiv 2024.3 ] | LaViLa† | GPT-4 | 62.4 |
| VideoAgent [13][ECCV 2024 ] | Video-LLava [32] | GPT-4 | 60.2 |
| | TOPA | CLIP-ViT-L | Llama2-7B | 41.2 |
| *(Our Text-Only Pre-Alignment)* | TOPA | CLIP-ViT-L | Llama3-8B | 44.2 |
| | TOPA | CLIP-ViT-L | Llama2-13B | 51.0 |

**Discussion 2: Video agents versus end-to-end video-LLM modeling.** Video agents have shown impressive results on the EgoSchema benchmark, aided by advanced LLMs and VLMs. However, a significant limitation of these approaches is their heavy reliance on the powerful LLMs. For example, the accuracy of Vamos drops by -11.6% when the GPT-4 is replaced with Llama2-13B, largely falling behind the performance of the TOPA-Llama2-13B model. The reliance on powerful closed-source LLMs restricts its application fields and introduces external overheads. Moreover, video agents make decisions based on the language format clues collected by VLMs. Converting the video content into language clues may lead to a limited upper bound compared to end-to-end modeling. Additionally, the inference speed of these approaches is another concern, since it involves multiple interactions with both VLMs and LLMs. In contrast, end-to-end video-LLM models, which condense the video into a sequence of embeddings as the input of LLM, are more efficient.

### 4.1.2 Zero-shot Results on NExT-QA, STAR and TVQA

Table 2 shows the multi-choice video QA results across various benchmarks. TOPA achieves impressive performance on the TVQA and EgoSchema benchmarks, significantly outperforming previous video pre-training models and image-to-video adaptation approaches. This indicates that our TOPA framework effectively enables LLMs to handle video input, despite not being pre-trained on real videos. However, for the NeXT-QA and STAR benchmarks, TOPA underperforms compared to SeViLA and IG-VLM. A major reason is that these benchmarks involve many fine-grained visual questions, including those about object locations and relationships. SeViLA and IG-VLM, benefiting from the advanced image-understanding capabilities of pre-trained VLMs such as LLaVA, excel in answering these fine-grained visual questions. In contrast, our TOPA framework primarily focuses on high-level semantic alignment. Moreover, during zero-shot inference, we project the visual features into the text feature space to bridge the modality gap, as described in Eq. 2. This cross-modal semantic projection process tends to overlook fine-grained visual details, such as object locations, which leads

to inferior performance on the STAR benchmark. We provide extensive qualitative results to illustrate TOPA's advantages and limitations across various video understanding tasks in Appendix A.3.

Table 2: Zero-shot results on multi-choice video QA benchmarks.

| Model (# Frames) | NExT-QA | | | | STAR | | | | | TVQA | EgoSchema |
|---|---|---|---|---|---|---|---|---|---|---|---|
| | Tem. | Cau. | Des. | **Avg.** | Int. | Seq. | Pre. | Fea. | **Avg.** | | |
| FrozenBiLM (10) [77] | - | - | - | - | - | - | - | - | - | 29.7 | 26.9 |
| InternVideo (8) [64] | 43.4 | 48.0 | 65.1 | 49.1 | 43.8 | 43.2 | 42.3 | 37.4 | 41.6 | 35.9 | 32.1 |
| SEVILA (32 → 4) [81] | 61.3 | 61.5 | 75.6 | 63.6 | 48.3 | 45.0 | 44.4 | 40.8 | 44.6 | 38.2 | 22.7 |
| IG-VLM-Llava7B (6) [22] | 63.1 | 57.3 | 74.9 | 63.1 | 49.3 | 50.1 | 48.4 | 48.8 | 49.6 | 42.1 | 35.8 |
| IG-VLM-Llava13B (6) [22] | 61.6 | 55.7 | 70.8 | 61.2 | 51.5 | 52.0 | 51.0 | 51.8 | 51.7 | 44.5 | 47.0 |
| TOPA-Llama2-7B (10) | 53.4 | 61.3 | 68.3 | 59.9 | 36.4 | 45.6 | 39.3 | 36.3 | 41.3 | 48.2 | 41.2 |
| TOPA-Llama3-8B (10) | 53.0 | 61.9 | 64.5 | 59.5 | 40.8 | 43.1 | 39.4 | 34.5 | 41.4 | 48.5 | 44.2 |
| TOPA-Llama2-13B (10) | 57.2 | 63.6 | 68.9 | 62.1 | 41.6 | 46.2 | 44.2 | 36.7 | 43.0 | 50.2 | 51.0 |

### 4.1.3 Results on MVBench

MVBench [29] is a recent video-language understanding benchmark that covers 20 challenging video tasks, regrouped from existing video-language benchmarks. Table 3 shows the results. TOPA demonstrates impressive results compared to previous image MLLM and video MLLM. It excels particularly in tasks requiring high-level video-language understanding, such as Scene Transition (ST), Episodic Reasoning (ER), and Unexpected Action (UA). TOPA Surprisingly excels in the Action Localization (AL) task, which requires identifying the moment an action occurs. This indicates that the text-only pre-alignment enables the LLM to understand temporal visual sequences. However, TOPA struggles with tasks that demand fine-grained visual understanding, such as Moving Direction (MR), Action Antonym (AA), and Object Shuffle (OS). A common challenge in these tasks is the requirement for detailed visual understanding. For example, Action Antonym involves identifying the direction of an action, while Object Shuffle involves locating objects. TOPA struggles in these fine-grained visual tasks since it is trained with CLIP text features. The modality gap between CLIP text features and image features hinders TOPA from capturing visual details. Further video instruction tuning might address this limitation, which we leave for future work. We provide qualitative results in Appendix A.3 to illustrate TOPA's advantages and limitations on various video understanding tasks.

Table 3: Evaluation results on MVBench. The results of other approaches are sourced from [29]. We gray out the results of VideoChat2 since it utilizes extensive annotated downstream video data.

| Model | LLM | Avg | AS | AP | AA | FA | UA | OE | OI | OS | MD | AL | ST | AC | MC | MA | SC | FP | CO | EN | ER | CI |
|---|---|---|---|---|---|---|---|---|---|---|---|---|---|---|---|---|---|---|---|---|---|---|
| Random | - | 27.3 | 25.0 | 25.0 | 33.3 | 25.0 | 25.0 | 33.3 | 25.0 | 33.3 | 25.0 | 25.0 | 25.0 | 33.3 | 25.0 | 33.3 | 33.3 | 25.0 | 33.3 | 25.0 | 20.0 | 30.9 |
| *Image MLLMs: Following [11], all models take 4 frames as input, with the output embeddings concatenated before feeding into the LLM.* | | | | | | | | | | | | | | | | | | | | | | |
| mPLUG-Owl-I [74] | LLaMA-7B | 29.4 | 25.0 | 20.0 | 44.5 | 27.0 | 23.5 | 36.0 | 24.0 | 34.0 | 23.0 | 24.0 | 34.5 | 34.5 | 22.0 | 31.5 | 40.0 | 24.0 | 37.0 | 25.5 | 21.0 | 37.0 |
| BLIP2 [27] | FlanT5-XL | 31.4 | 24.5 | 29.0 | 33.5 | 17.0 | 42.0 | 51.5 | 26.0 | 31.0 | 25.5 | 26.0 | 32.5 | 25.5 | 30.0 | 40.0 | 42.0 | 27.0 | 30.0 | 26.0 | 37.0 | 31.0 |
| LLaMA-Adapter [84] | LLaMA-7B | 31.7 | 23.0 | 28.0 | 51.0 | 30.0 | 33.0 | 53.5 | 32.5 | 33.5 | 25.5 | 21.5 | 30.5 | 29.0 | 22.5 | 41.5 | 39.5 | 25.0 | 31.5 | 22.5 | 28.0 | 32.0 |
| Otter-I [25] | MPT-7B | 33.5 | 34.5 | 32.0 | 39.5 | 30.5 | 38.5 | 48.5 | 44.0 | 29.5 | 19.0 | 25.5 | 55.0 | 20.0 | 32.5 | 28.5 | 39.0 | 28.0 | 27.0 | 32.0 | 29.0 | 36.5 |
| MiniGPT-4 [89] | Vicuna-7B | 18.8 | 16.0 | 18.0 | 26.0 | 21.5 | 16.0 | 29.5 | 25.5 | 13.0 | 11.5 | 12.0 | 9.5 | 32.5 | 15.5 | 8.0 | 34.0 | 26.0 | 29.5 | 19.0 | 9.9 | 3.0 |
| InstructBLIP [11] | Vicuna-7B | 32.5 | 20.0 | 16.5 | 46.0 | 24.5 | 46.0 | 51.0 | 26.0 | 37.5 | 22.0 | 23.0 | 46.5 | 42.5 | 26.5 | 40.5 | 32.0 | 25.5 | 30.0 | 25.5 | 30.5 | 38.0 |
| LLaVA [34] | Vicuna-7B | 36.0 | 28.0 | 39.5 | 63.0 | 30.5 | 39.0 | 53.0 | 41.0 | 41.5 | 23.0 | 20.5 | 45.0 | 34.0 | 20.5 | 38.5 | 47.0 | 25.0 | 36.0 | 27.0 | 26.5 | 42.0 |
| *Video MLLMs: All models take 16 frames as input* | | | | | | | | | | | | | | | | | | | | | | |
| Otter-V [25] | LLaMA-7B | 26.8 | 23.0 | 23.0 | 27.5 | 27.0 | 29.5 | 53.0 | 28.0 | 33.0 | 24.5 | 23.5 | 27.5 | 26.0 | 28.5 | 18.0 | 38.5 | 22.0 | 22.0 | 23.5 | 19.0 | 19.5 |
| mPLUG-Owl-V [79] | LLaMA-7B | 29.7 | 22.0 | 28.0 | 34.0 | 29.0 | 29.0 | 40.5 | 27.0 | 31.5 | 27.0 | 23.0 | 29.0 | 31.5 | 27.0 | 40.0 | 44.0 | 24.0 | 31.0 | 26.0 | 20.5 | 29.5 |
| *Instructed Video MLLMs: All models take 16 frames as input, with the exception of VideoChatGPT, which uses 100 frames.* | | | | | | | | | | | | | | | | | | | | | | |
| VideoChatGPT [38] | Vicuna-7B | 32.7 | 23.5 | 26.0 | 62.0 | 22.5 | 26.5 | 54.0 | 28.0 | 40.0 | 23.0 | 20.0 | 31.0 | 30.5 | 25.5 | 39.5 | 48.5 | 29.0 | 33.0 | 29.5 | 26.0 | 35.5 |
| VideoLLaMA [83] | Vicuna-7B | 34.1 | 27.5 | 25.5 | 51.0 | 29.0 | 39.0 | 48.0 | 40.5 | 38.0 | 22.5 | 22.5 | 43.0 | 34.0 | 22.5 | 32.5 | 45.5 | 32.5 | 40.0 | 30.0 | 21.0 | 37.0 |
| VideoChat [28] | Vicuna-7B | 35.5 | 33.5 | 26.5 | 56.0 | 33.5 | 40.5 | 53.0 | 40.5 | 30.0 | 25.5 | 27.0 | 48.5 | 35.0 | 20.5 | 42.5 | 46.0 | 26.5 | 41.0 | 23.5 | 23.5 | 36.0 |
| VideoChat2 [29] | Vicuna-7B | 51.1 | 66.0 | 47.5 | 83.5 | 49.5 | 60.0 | 58.0 | 71.5 | 42.5 | 23.0 | 23.0 | 88.5 | 39.0 | 42.0 | 58.5 | 44.0 | 49.0 | 36.5 | 35.0 | 40.5 | 65.5 |
| *Text-Only Pre-Alignment Video-MLLM: TOPA zero-shot inference with 10 frames.* | | | | | | | | | | | | | | | | | | | | | | |
| TOPA-ZeroShot | LLama2-7B | 39.8 | 42.0 | 38.5 | 35.0 | 34.5 | 66.0 | 52.5 | 47.5 | 28.0 | 22.0 | 37.5 | 81.0 | 38.0 | 24.0 | 42.5 | 41.5 | 28.5 | 34.0 | 23.5 | 49.0 | 30.5 |
| TOPA-ZeroShot | LLama2-13B | 42.5 | 38.0 | 40.0 | 42.5 | 35.0 | 69.0 | 52.5 | 58.5 | 29.5 | 22.5 | 43.5 | 80.5 | 38.0 | 25.5 | 43.0 | 43.0 | 29.5 | 37.5 | 38.5 | 50.0 | 32.5 |

| AS | AP | AA | FA | UA | OE | OI | OS | MD | AL |
|---|---|---|---|---|---|---|---|---|---|
| Action Sequence | Action Prediction | Action Antonym | Fine-grained Action | Unexpected Action | Object Existence | Object Interaction | Object Shuffle | Moving Direction | Action Localization |
| STAR [70] | STAR [70] | PAXION [68] | MiT V1 [43] | FunQA [73] | CLEVRER [80] | STAR [70] | Perception Test [46] | CLEVRER [80] | Charades-STA [14] |
| **ST** | **AC** | **MC** | **MA** | **SC** | **FP** | **CO** | **EN** | **ER** | **CI** |
| Scene Transition | Action Counting | Moving Counting | Moving Attribute | State Change | Fine-grained Pose | Character Order | Egocentric Navigation | Episodic Reasoning | Counterfactual Inference |
| MovieNet [18] | Perception Test [46] | CLEVRER [80] | CLEVRER [80] | Perception Test [46] | NTU RGB+D [35] | Perception Test [46] | VLN-CE [69] | TVQA [24] | CLEVRER [80] |

## 4.2 Supervised Finetuning

In this section, we further finetune the pre-aligned TOPA models to study the benefits of TOPA for downstream supervised learning. During finetuning, TOPA directly takes the video feature as input without the cross-modal projection. More finetuning details for each dataset are provided in Appendix E.2. Table 4 shows the finetuning results on multi-choice video QA dataset. For comparison, we include baseline models without text-only pretraining. Our text-only pre-alignment consistently improves the performance across three benchmarks. Notably, TOPA-Llama2-7B achieves 67.1% accuracy on TVQA, outperforming other approaches by a large margin. These results suggest that our text-only pre-alignment, even without training on real videos, has a similar effect to conventional video-language pre-training.

Table 4: Finetuning results on NExT-QA, STAR and TVQA.

| Model (# Frames) | NExT-QA | | | | STAR | | | | | TVQA |
| | Tem. | Cau. | Des. | **Avg.** | Int. | Seq. | Pre. | Fea. | **Avg.** | |
|---|---|---|---|---|---|---|---|---|---|---|
| FrozenBiLM (10) [77] | - | - | - | - | - | - | - | - | - | 57.5 |
| InternVideo (8) [64] | 58.5 | 62.5 | 75.8 | 63.2 | 62.7 | 65.6 | 54.9 | 51.9 | 58.7 | 57.2 |
| BLIP-2$^{voting}$ (4) [81] | 65.2 | 70.1 | 80.1 | 70.1 | 52.3 | 54.8 | 49.0 | 51.2 | 51.8 | 54.5 |
| SEVILA (32 → 4) [81] | 69.4 | 74.2 | 81.3 | 73.8 | 63.7 | 70.4 | 63.1 | 62.4 | 64.9 | 61.6 |
| Llama-VQA-7B (10) [23] | 69.2 | 72.7 | 75.8 | 72.0 | 66.2 | 67.9 | 57.2 | 52.7 | 65.4 | - |
| Baseline (10) | 65.3 | 69.0 | 72.6 | 68.4 | 60.8 | 61.5 | 49.2 | 49.8 | 59.4 | 63.8 |
| TOPA-Llama2-7B (10) | 71.3 | 74.2 | 78.5 | 73.9 | 66.8 | 68.9 | 59.1 | 55.5 | 66.4 | 67.1 |
| | +6.0 | +5.2 | +5.9 | +5.5 | +6.0 | +7.4 | +9.9 | +5.7 | +7.0 | +3.3 |
| Baseline (10) | 66.0 | 69.7 | 73.7 | 69.1 | 61.4 | 62.4 | 50.6 | 51.8 | 60.3 | 66.2 |
| TOPA-Llama3-8B (10) | 70.1 | 74.5 | 74.6 | 73.1 | 66.3 | 67.0 | 59.1 | 56.5 | 65.4 | 68.1 |
| | +4.1 | +4.8 | +0.9 | +4.0 | +4.9 | +4.6 | +8.5 | +4.7 | +5.1 | +1.9 |
| Baseline (10) | 67.8 | 71.6 | 75.2 | 70.9 | 58.7 | 59.5 | 54.3 | 51.8 | 58.2 | 66.6 |
| TOPA-Llama2-13B (10) | 72.1 | 75.8 | 79.3 | 75.1 | 66.8 | 68.3 | 61.0 | 55.1 | 66.3 | 69.0 |
| | +4.3 | +4.2 | +4.1 | +4.2 | +8.1 | +8.8 | +6.7 | +3.3 | +8.1 | +2.4 |

**Data-efficient finetuning.** Figure 3 shows the results of finetuning LLMs with various ratios of training data. TOPA trained with 10% data achieves 64.7% Top 1 accuracy on NeXT-QA benchmark, significantly outperforming the baseline that without text-only pre-alignment. Besides, when trained with less than 20% data, the baseline model even performs worse than TOPA-zeroshot on NeXT-QA and TVQA, clearly demonstrating the effectiveness of TOPA in limited annotated data scenarios.

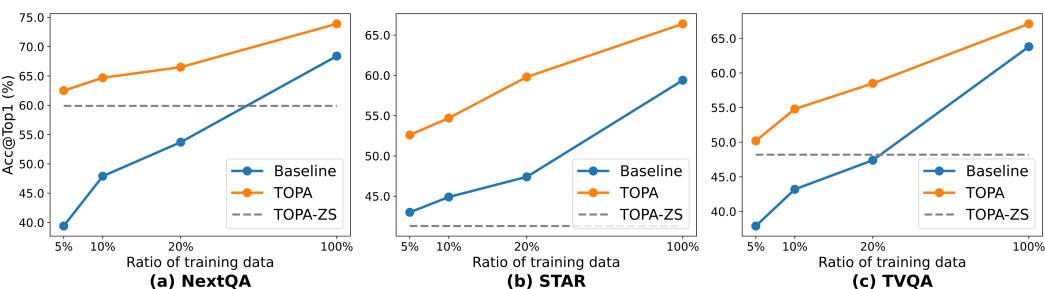

Figure 3: Results of finetuning TOPA with various ratios of training data.

## 4.3 Video Captioning

**Results on zero-shot video captioning.** We further perform zero-shot video captioning on MSR-VTT [75] and VATEX [62]. As shown in Table 5, TOPA largely outperforms previous text-only approaches like Decap which is trained on captions sourced from CC3M [53]. TOPA even outperforms the video-text pre-training approaches like VideoCoCa, which is pre-trained on millions of videos-text data, demonstrating that TOPA is an efficient and effective framework for video-LLM alignment.

Table 5: Zero-shot video captioning results. We report CIDEr score for all benchmarks. *VT* denotes ⟨*video clip, text*⟩ pairs, *IT* denotes ⟨*image, text*⟩ pairs, and *WP* denotes webpages consisting of interleaved image and text data.

| | Training data | MSR-VTT | VATEX |
|---|---|---|---|
| *(Web video-text Pre-training)* | | | |
| VideoCoCa-g [76] | 144M *VT* | 27.1 | 22.8 |
| Flamingo-3B [1] | 27M *VT* & 2.1B *IT* & 43M *WP* | - | 40.1 |
| Flamingo-9B [1] | 27M *VT* & 2.1B *IT* & 43M *WP* | - | 39.5 |
| VideoPrism-B [86] w/ PaLM-2-1B | 618M *VT* | 40.3 | 24.2 |
| VideoPrism-B [86] w/ PaLM-2-8B | 618M *VT* | 38.5 | 31.7 |
| *(Text-only Pre-training)* | | | |
| DeCap [30] | 3M *Captions* | 18.6 | 18.7 |
| TOPA-Llama2-7B | 721K *TextVid* | 32.9 | 31.0 |
| TOPA-Llama2-13B | 721K *TextVid* | 33.4 | 32.0 |

## 4.4 Ablations

**LLM prior in video-language understanding.** To investigate the impact of LLM prior in multi-choice video QA, we conduct experiments on EgoSchema with the blind setting, where only the questions and choices are provided to the LLM. Table 6 shows the results. Bard and GPT-4-Turbo achieve 33.2% and 30.8% accuracy, respectively. Gemini-Pro-1.0 reaches 38.2% accuracy. These blind results of advanced LLMs suggest that in some video QA cases, LLMs can accurately choose the correct answer solely based on the question and choices, without visual input. However, the blind performance of Llama2-7B and Llama2-13B is inferior, potentially due to their smaller model size. After training on the TextVid dataset, TOPA-Llama2-13B achieves a blind accuracy of 37.5% (or +11.7%), closely approaching that of Gemini-Pro-1.0 model. These results suggest that text-only pre-alignment can effectively prepare LLMs for downstream video-language tasks by leveraging specialized text-only tasks, even in complex scenarios where the original LLMs are limited.

Table 6: Blind results on EgoSchema. † denotes results sourced from [4].

| | Visual Input | ES Full |
|---|---|---|
| Random Selection | | 20.0 |
| GPT-4-Turbo † | ✗ | 30.8 |
| Bard † | | 33.2 |
| Gemini-Pro-1.0 | | 38.2 |
| Llama2-7B | ✗ | 20.1 |
| Llama2-13B | | 25.8 |
| TOPA-Llama2-7B | ✗ | 29.3 |
| TOPA-Llama2-13B | | 37.5 |
| TOPA-Llama2-7B | ✔ | 41.2 |
| TOPA-Llama2-13B | | 51.0 |

Table 7: Ablation on video frames.

| TOPA | #Frame | NextQA | ES Full |
|---|---|---|---|
| | 1 | 56.1 | 39.4 |
| Llama2-7B | 5 | 58.9 (+2.8) | 41.0 (+1.6) |
| | 10 | 59.9 (+3.8) | 41.2 (+1.8) |
| | 1 | 57.3 | 47.6 |
| Llama2-13B | 5 | 60.8 (+3.5) | 50.5 (+2.9) |
| | 10 | 62.1 (+4.8) | 51.0 (+3.4) |

**The impact of video frames.** To better investigate TOPA's capability in understanding temporal dynamics of real videos, we conduct experiments with different number of frames. Table 7 shows the results. Multiple frames input consistently enhances performance on NeXT-QA and EgoSchema for both TOPA-Llama2-7B and TOPA-Llama2-13B. This indicates that the text-only pre-alignment effectively enables the LLM to handle multiple video frames, despite not being trained on real videos.

## 5 Conclusions

In this paper, we introduce TOPA, a text-only pre-alignment framework designed for aligning LLMs with video modality without requiring training on real videos. TOPA has demonstrated remarkable performance on the recent, challenging long-form video understanding benchmark, *i.e.*, EgoSchema, showcasing that a text-only approach is effective in capturing the dynamics of long-form videos. Our approach, which includes data generation and text-only pre-alignment, has potential applications across various vision-language tasks where obtaining paired vision-language data is difficult.

## Acknowledgements

This work was supported by National Key R&D Program of China (No. 2023YFC3305600), the National Natural Science Foundation of China (U2336212), the Fundamental Research Funds for the Zhejiang Provincial Universities (226-2024-00208), Lu's Graduate Education International Exchange Foundation and the National Research Foundation, Singapore under its Strategic Capability Research Centres Funding Initiative. Any opinions, findings and conclusions or recommendations expressed in this material are those of the author(s) and do not reflect the views of National Research Foundation, Singapore. The computational work was partially performed on resources of the National Supercomputing Centre, Singapore (https://www.nscc.sg).

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

# Appendix

In Appendix A, we provide additional experiments and analysis.

- In Appendix A.1, we further discuss the Multi-choice QA task and study the impact of the multi-choice Tideo QA pre-training.
- In Appendix A.2, we study the impact of cross-modal projection (Eq. 2).
- In Appendix A.3, we provide extensive qualitative results to illustrate TOPA's advantages and limitations across various video understanding tasks.

Appendix B: The limitations of TOPA.

Appendix C: The broader impact of TOPA.

Appendix D: The details of proposed TextVid dataset.

Appendix E.1, The details of benchmarks.

Appendix E.2, The training details of TOPA.

Appendix F: The prompts used in this paper.

Appendix G: The licenses of datasets, codes and models used in this paper.

Appendix H: Examples from TextVid.

# A   Additional Experiments

## A.1   Further Discussion on Multi-Choice Video QA Task

Table 8: Multi-choice video QA on EgoSchema subset and full set. "Gap" refers to the difference in performance between the subset and the full set

| Method | Eval Mode | ES Subset | ES Full | Gap |
|---|---|---|---|---|
| LongViViT [45] | Similarity | 56.8 | 33.3 | -23.5 |
| MC-ViT-L [4] | Similarity | 62.6 | 44.0 | -18.6 |
| MVU [49] | LLM logits | 60.3 | 37.6 | -22.7 |
| LangRepo-Mixtral-8×7B-(12B active) [21] | LLM logits | 66.2 | 41.2 | -25.0 |
| VideoAgent (GPT-4) [63] | LLM Selection | 60.2 | 54.1 | -6.1 |
| TOPA-LLama2-13B | LLM Logits | 67.5 | 41.6 | -25.9 |
| TOPA-LLama2-13B | LLM Selection | 51.2 | 51.0 | -0.2 |
| TOPA-LLama2-7B | LLM Logits | 64.5 | 41.7 | -22.8 |
| TOPA-LLama2-7B | LLM Selection | 40.4 | 41.2 | +0.8 |
| TOPA-LLama2-7B (w/o multi-choice training) | LLM Logits | 65.1 | 40.5 | -24.6 |
| TOPA-LLama2-7B (w/o multi-choice training) | LLM Selection | 24.3 | 24.7 | +0.4 |

A significant advantage of the text-only framework is that we can utilize the LLM to automatically generate diverse language-based supervisions as needed, such as the multi-choice QA pairs. To explore the impact of the multi-choice QA training tasks, we conduct an ablation study as shown in Table 8. We would like to first introduce the different evaluation modes for multi-choice video QA tasks: (1) **LLM Selection**: Asking the LLM to predict the correct answer given the video-question-choices. (2) **LLM Logits**: Given the video and question as LLM context, we calculate the logits for each choice by averaging the logits of all words within the choice. The choice with higher logit tends to match the video-question context better and is thus selected as the predicted answer. (3) **Similarity Comparison** [45, 4]: Mapping the multiple question-choice pairs and video to a common feature space and calculating the similarity between the video and each question-choice.

**The performance gap between the EgoSchema subset and full set.** Previous work [4, 45] highlights a huge performance gap between the subset and the full set of EgoSchema as shown in Table 8. While concurrent work [21, 49] introduces log-likelihood based approaches for LLM inference, which significantly improve the performance on EgoSchema subset, the issue of the performance gap still persists. In this paper, we observe that such a performance gap phenomenon also occurs in approaches based on LLM logits. However, but it diminishes or even disappears in methods employing LLM selection. We find that this phenomenon may be attributed to differences in the linguistic structures of the choices, as shown below. The choices in the subset often differ in several key works like "create", "repair" and "clean". The similarity or logit can effectively identify this keyword-level difference to select a more appropriate choice. Conversely, the choices in the full set display more substantial linguistic differences. These variations introduce significant language biases, *i.e.*, some sentences naturally receive higher logits in LLM, complicating the reliance on similarity or logit for choice selection. In contrast, LLM selection methods take all the choices within the context, allowing the LLM to leverage its robust contextual understanding to select the correct choice.

Question-Choices examples from subset:
*Q: Can you summarize the primary objective and the steps the person took throughout the video to achieve it? ensure your answer captures the essence of the video without listing all actions.*
*A: The main aim of the person's primary objective was to **create and build** a new, sturdy wooden bench.*
*B: The primary objective for the person was to thoroughly **repair and restore** the wooden bench.*
*C: The person's primary objective was to thoroughly **clean and sanitize** the wooden bench's surface.*

Question-Choices examples from full set:
*Q: Considering the entire video, what would you identify as the most crucial moments in the person's shopping experience and why?*
*A: Following a strict shopping list as a guideline and rejecting unfit produce.*
*B: Conducting taste tests and checking for the freshness of each vegetable.*
*C: Using math algorithm for optimal vegetable selection.*

**LLM for multi-choice QA.** In Table 8, we observe a notable phenomenon where the TOPA models achieve impressive results on the subset with the logits evaluation mode. TOPA-LLama2-13B achieves 67.5% top1 accuracy, surpassing GPT-4-based video agents. However, when evaluated with the multi-choice selection mode, the performance of the subset declines to 51.2%, but the performance of the full set increases from 41.6% to 51.0%. These results suggest that while the LLM is capable of selecting the answer from multiple choices, it is less sensitive to the keywords within those choices. In contrast, the logit-based approach is sensitive to the keywords but has difficulty with complex sentence understanding.

**The impact of the Multi-Choice Tideo QA pre-training.** In Table 8, we report the results of TOPA without the multi-choice Tideo QA task, *i.e.*, trained with Tideo summarization and Tideo QA tasks. In this case, we find that TOPA-LLama2-7B maintains similar performance when evaluated with the logit mode. However, there is a significant performance drop when evaluated with the multi-choice selection mode. This result suggests that while the LLM is adapted to process video inputs, its capability is somewhat constrained and can not extend to more complex video-language tasks beyond the pre-training tasks. This finding highlights the advantage of our text-only data generation and text-only pre-alignment framework, which enable us to develop a variety of pre-alignment tasks to better equip the LLM for general video-language tasks such as dense captioning, multi-choice video QA, and video chat.

## A.2    The CLIP Modality Gap

TOPA is pretrained with CLIP text features while inferenced with CLIP image features. We employ a modality projection approach, *i.e.*, Eq. 2, to bridge this CLIP modality gap during zero-shot inference. Table 9 shows the impact of Eq. 2. TOPA shows inferior results when directly taking the visual feature as input due to the modality gap problem. The projection approach effectively alleviates such a modality gap problem without additional training.

Table 9: Ablation on the modality projection (Eq. 2). Results on EgoSchema full set.

| Model | without Eq. 2 | with Eq. 2 |
|---|---|---|
| TOPA-LLama2-7B | 30.6 | 41.2 |
| TOPA-LLama2-13B | 38.3 | 51.0 |

## A.3 Qualitative Results and Analysis

We present qualitative results to illustrate the capabilities and limitations of TOPA across various video understanding tasks. Figure 4 shows qualitative results on the NExT-QA validation set. Figure 5 shows qualitative results on the EgoSchema subset. Figure 6 - 9 shows qualitative results on 20 video understanding tasks from MVBench.

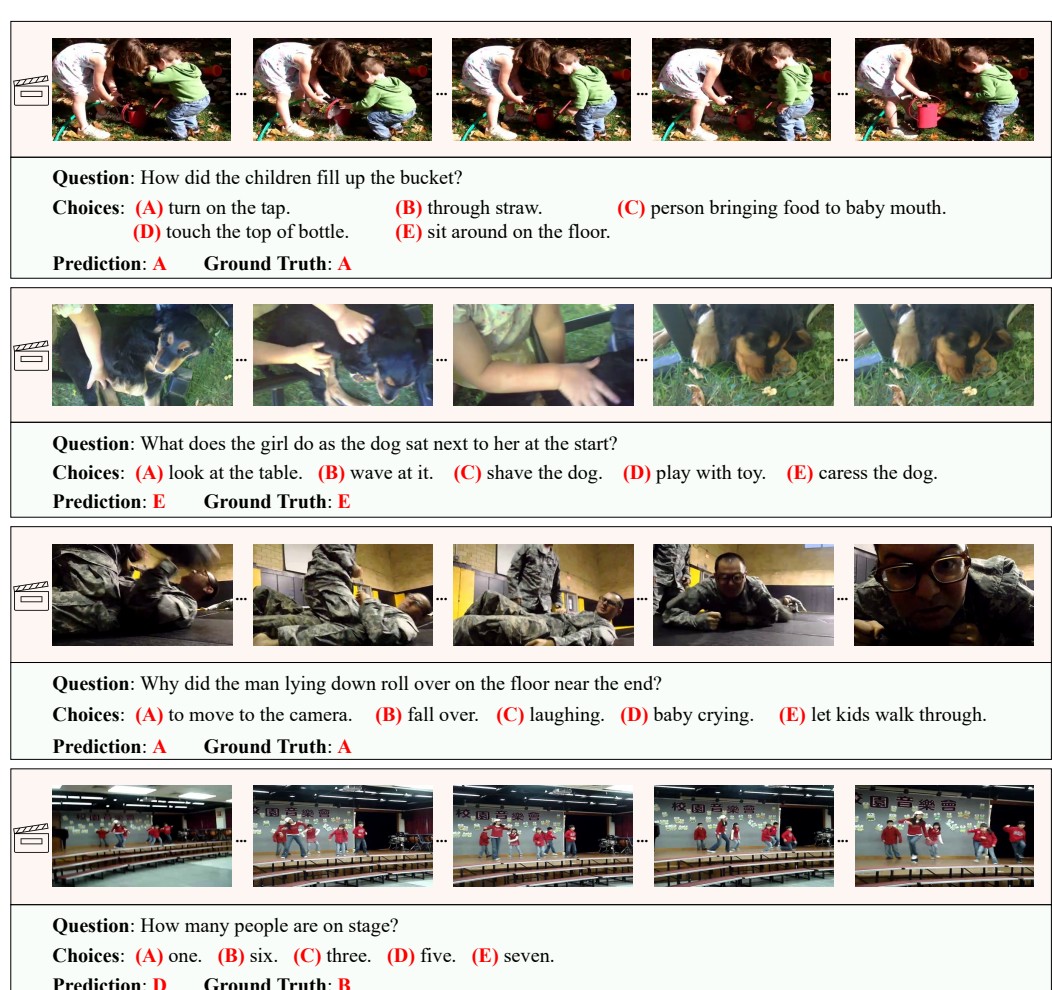

Figure 4: Qualitative results on NeXT-QA. TOPA effectively performs complex video understanding tasks. Additionally, a failure case is also shown in the figure, *i.e.*, in the last sample, TOPA failed to accurately count the number of people.

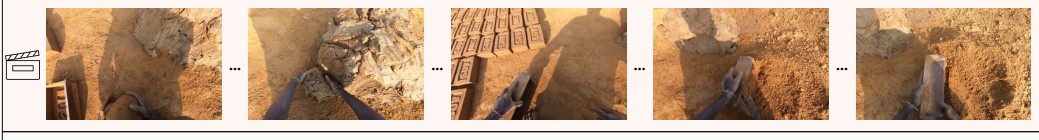

**Question**: What is the overall process and purpose of the actions performed by the person in the video?

**Choices**:

**(A)** The person is molding bricks with mortar and wet clay.

**(B)** Currently, the person is diligently working on constructing a wall.

**(C)** Currently, artist the person is diligently working on creating a unique sculpture.

**(D)** Currently, young the person is actively playing and having fun with mud.

**(E)** The person is doing a science experiment.

**Prediction**: **A**     **Ground Truth**: **A**

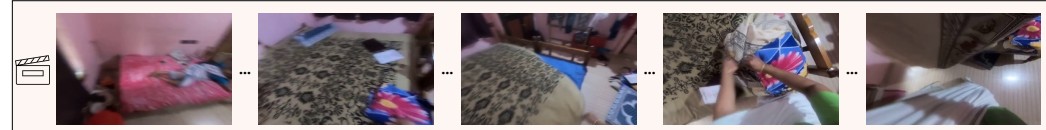

**Question**: How does the person interact with the camera during the video and what might be the reason behind these adjustments?

**Choices**:

**(A)** Casually, the person adjusts the camera angle skillfully to achieve a more visually appealing view of herself.

**(B)** The person skillfully adjusts the camera angle to achieve a better, enhanced view of the entire room.

**(C)** Casually, the person adjusts the camera angle to get a more improved, better view of the entrance door.

**(D)** The person adjusts the camera to get a better view of the window.

**(E)** The person adjusts the camera to get a better view of the bed and the cloths.

**Prediction**: **E**     **Ground Truth**: **E**

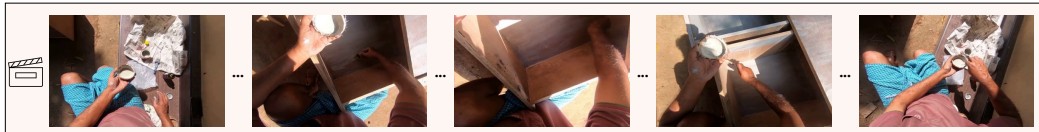

**Question**: Determine the primary purpose of the person's actions in the video, and explain the importance of the repetitive actions involved in this process.

**Choices**:

**(A)** In the kitchen, the person is diligently making a cup of coffee. the repetitive actions, or steps, are necessary to ensure that the coffee concoction is mixed evenly throughout.

**(B)** The person is stirring a pot of soup. the repetitive actions are necessary to ensure that the soup is cooked evenly.

**(C)** The person is painting a piece of furniture. the repetitive actions are necessary to ensure that the paint is applied evenly.

**(D)** The person is diligently washing dishes. the repetitive actions performed are extremely necessary to ensure that every single dish is thoroughly clean.

**(E)** Currently, the person is diligently brushing his teeth. these repetitive actions are crucial and necessary to effectively ensure that his teeth remain ultimately clean and healthy.

**Prediction**: **C**     **Ground Truth**: **C**

Figure 5: EgoSchema presents unique challenges compared to previous video benchmarks. The questions in EgoSchema are complex and demand advanced video capabilities, encompassing both recognition and reasoning skills.

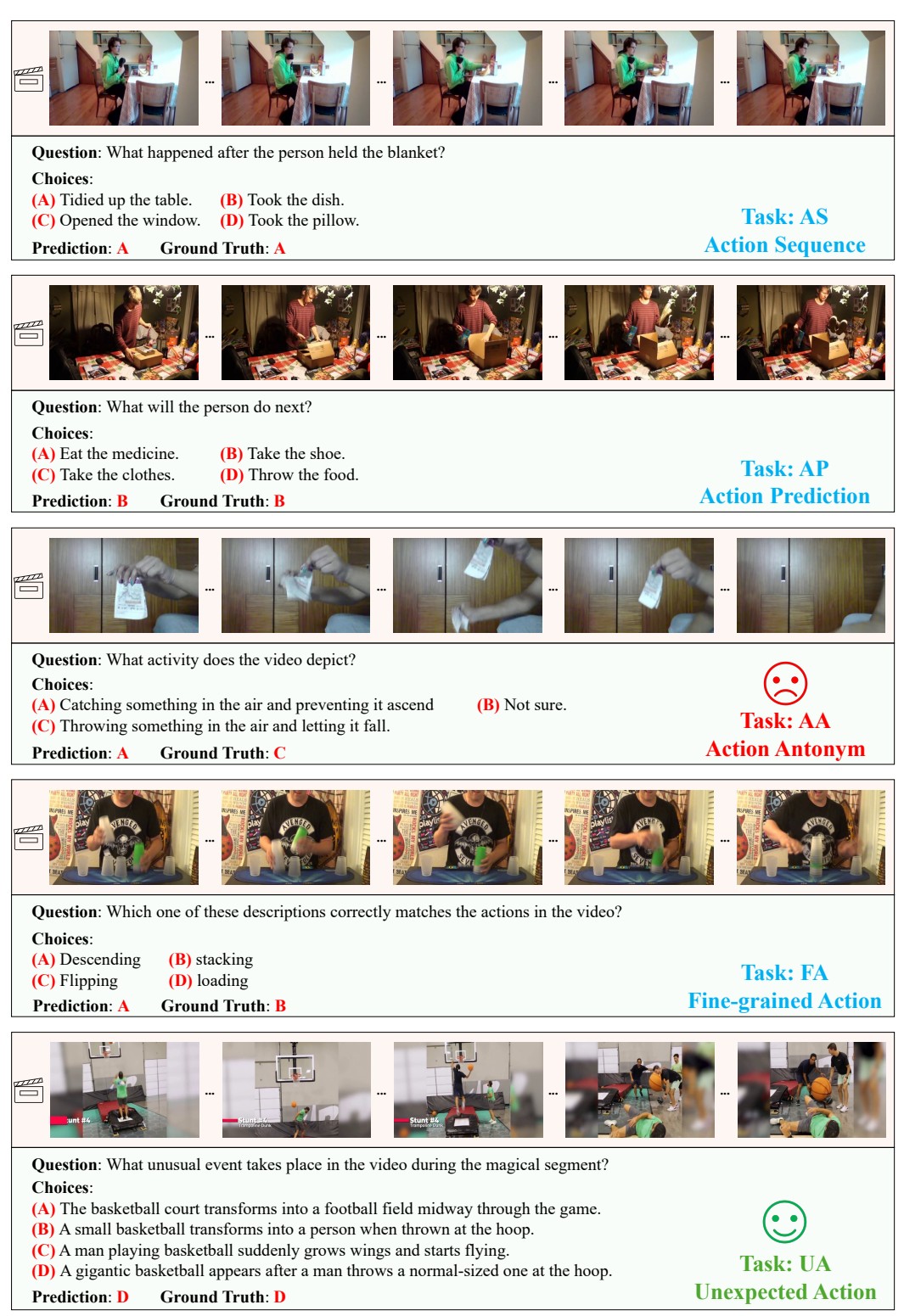

**Question**: What happened after the person held the blanket?
**Choices**:
(A) Tidied up the table.  (B) Took the dish.
(C) Opened the window.  (D) Took the pillow.

**Prediction**: A  **Ground Truth**: A

**Task: AS**
**Action Sequence**

**Question**: What will the person do next?
**Choices**:
(A) Eat the medicine.  (B) Take the shoe.
(C) Take the clothes.  (D) Throw the food.

**Prediction**: B  **Ground Truth**: B

**Task: AP**
**Action Prediction**

**Question**: What activity does the video depict?
**Choices**:
(A) Catching something in the air and preventing it ascend  (B) Not sure.
(C) Throwing something in the air and letting it fall.

**Prediction**: A  **Ground Truth**: C

**Task: AA**
**Action Antonym**

**Question**: Which one of these descriptions correctly matches the actions in the video?
**Choices**:
(A) Descending  (B) stacking
(C) Flipping  (D) loading

**Prediction**: A  **Ground Truth**: B

**Task: FA**
**Fine-grained Action**

**Question**: What unusual event takes place in the video during the magical segment?
**Choices**:
(A) The basketball court transforms into a football field midway through the game.
(B) A small basketball transforms into a person when thrown at the hoop.
(C) A man playing basketball suddenly grows wings and starts flying.
(D) A gigantic basketball appears after a man throws a normal-sized one at the hoop.

**Prediction**: D  **Ground Truth**: D

**Task: UA**
**Unexpected Action**

Figure 6: Qualitative results on MVBench (Task 1-5). The tasks where TOPA performs well, average, or poorly are marked in green, blue, and red colors respectively.

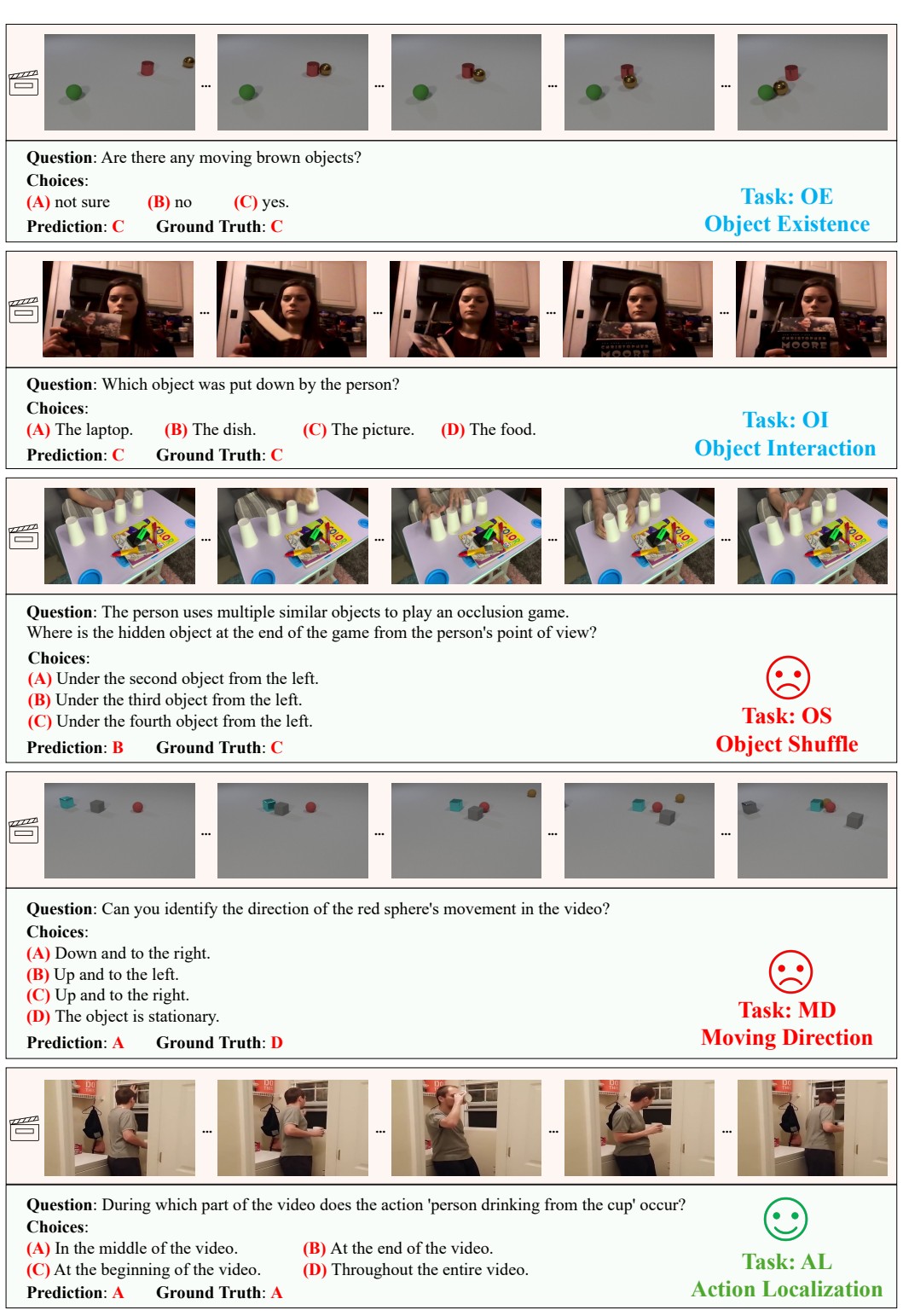

Figure 7: Qualitative results on MVBench (Task 6-10).

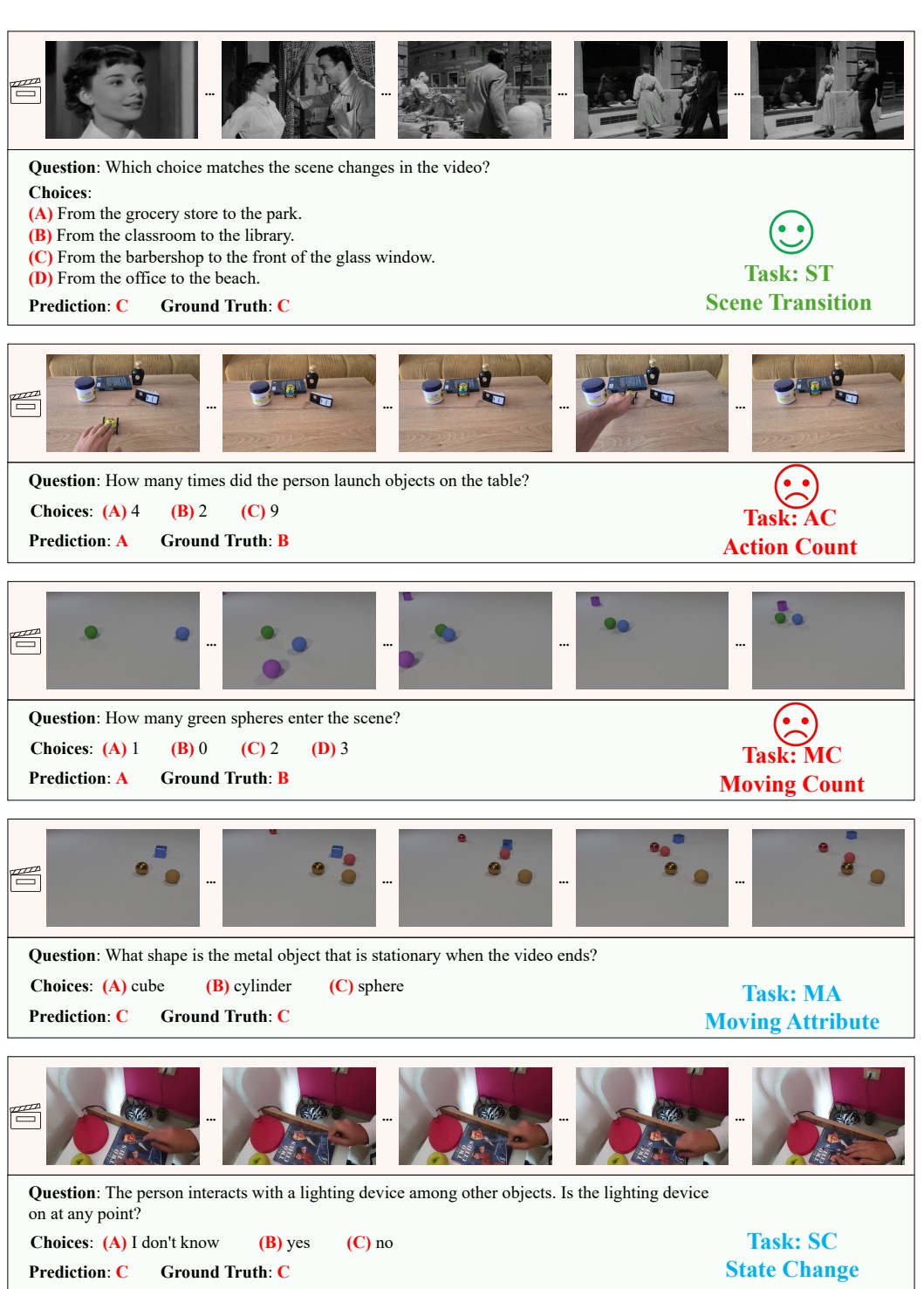

**Question**: Which choice matches the scene changes in the video?
**Choices**:
**(A)** From the grocery store to the park.
**(B)** From the classroom to the library.
**(C)** From the barbershop to the front of the glass window.
**(D)** From the office to the beach.

**Prediction**: **C**    **Ground Truth**: **C**

**Task: ST**
**Scene Transition**

**Question**: How many times did the person launch objects on the table?

**Choices**:  **(A)** 4    **(B)** 2    **(C)** 9

**Prediction**: **A**    **Ground Truth**: **B**

**Task: AC**
**Action Count**

**Question**: How many green spheres enter the scene?

**Choices**:  **(A)** 1    **(B)** 0    **(C)** 2    **(D)** 3

**Prediction**: **A**    **Ground Truth**: **B**

**Task: MC**
**Moving Count**

**Question**: What shape is the metal object that is stationary when the video ends?

**Choices**:  **(A)** cube    **(B)** cylinder    **(C)** sphere

**Prediction**: **C**    **Ground Truth**: **C**

**Task: MA**
**Moving Attribute**

**Question**: The person interacts with a lighting device among other objects. Is the lighting device on at any point?

**Choices**:  **(A)** I don't know    **(B)** yes    **(C)** no

**Prediction**: **C**    **Ground Truth**: **C**

**Task: SC**
**State Change**

Figure 8: Qualitative results on MVBench (Task 11-15).

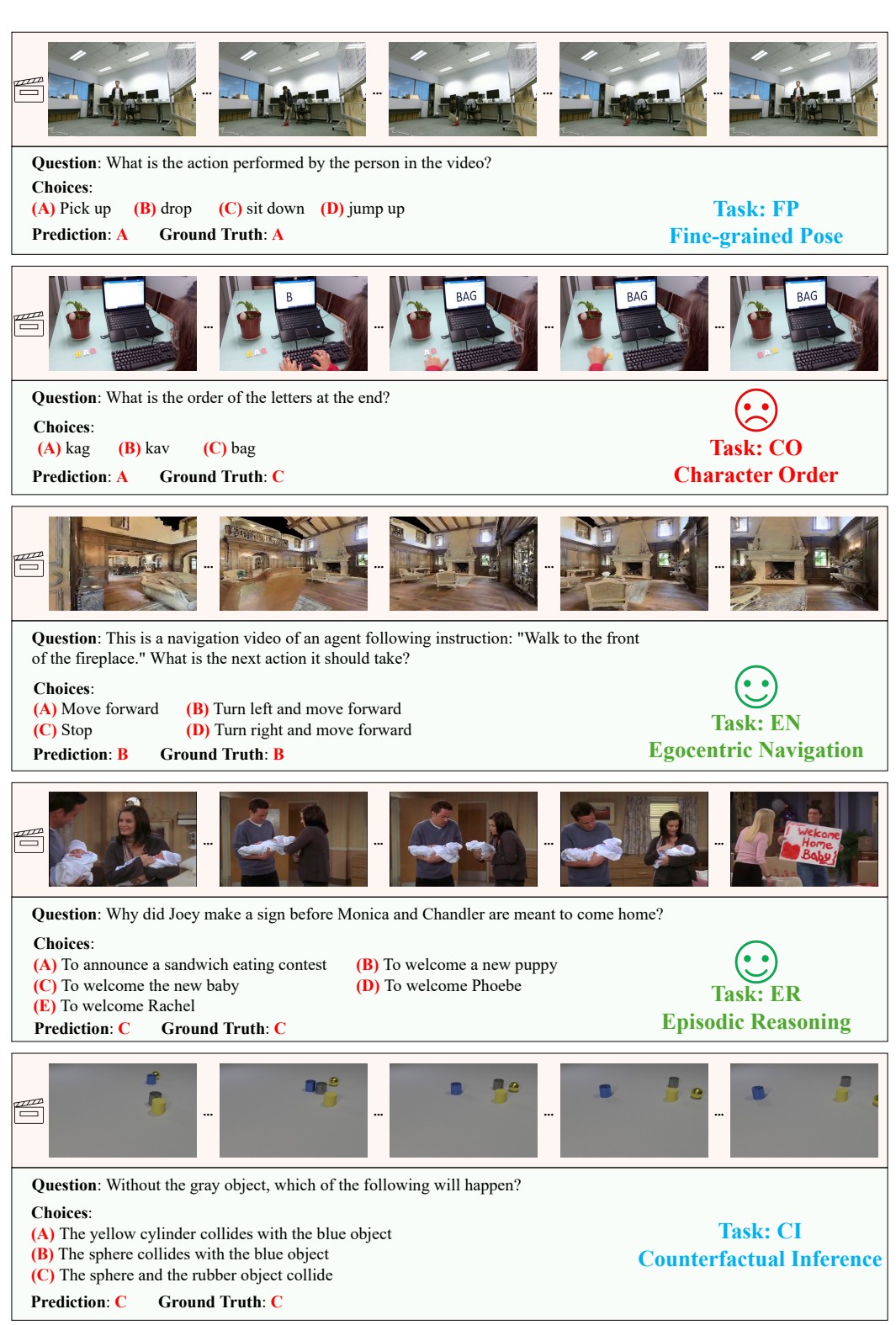

**Question**: What is the action performed by the person in the video?

**Choices**:

(A) Pick up    (B) drop    (C) sit down    (D) jump up

**Prediction**: A      **Ground Truth**: A

**Task: FP**
**Fine-grained Pose**

---

**Question**: What is the order of the letters at the end?

**Choices**:

(A) kag      (B) kav      (C) bag

**Prediction**: A      **Ground Truth**: C

**Task: CO**
**Character Order**

---

**Question**: This is a navigation video of an agent following instruction: "Walk to the front of the fireplace." What is the next action it should take?

**Choices**:

(A) Move forward      (B) Turn left and move forward

(C) Stop      (D) Turn right and move forward

**Prediction**: B      **Ground Truth**: B

**Task: EN**
**Egocentric Navigation**

---

**Question**: Why did Joey make a sign before Monica and Chandler are meant to come home?

**Choices**:

(A) To announce a sandwich eating contest      (B) To welcome a new puppy

(C) To welcome the new baby      (D) To welcome Phoebe

(E) To welcome Rachel

**Prediction**: C      **Ground Truth**: C

**Task: ER**
**Episodic Reasoning**

---

**Question**: Without the gray object, which of the following will happen?

**Choices**:

(A) The yellow cylinder collides with the blue object

(B) The sphere collides with the blue object

(C) The sphere and the rubber object collide

**Prediction**: C      **Ground Truth**: C

**Task: CI**
**Counterfactual Inference**

Figure 9: Qualitative results on MVBench (Task 15-20).

# B Limitations.

**Modality gap in CLIP.** Despite the fact that TOPA achieves impressive results, a significant limitation in TOPA is the gap between the CLIP text feature and CLIP image feature. On the one hand, we use the CLIP text feature for pre-alignment, while inference is with the CLIP visual feature. The modality gap makes the performance degrade, despite employing a modality projection mechanism to mitigate it. On the other hand, the CLIP text features cannot fully capture the fine-grained visual details present in actual images, such as object locations and relationships. This limitation causes TOPA to struggle in scenarios where questions involve detailed visual information, as shown in Appendix A.3.

**Struggles in fine-grained visual understanding.** In TOPA, we propose textual videos to mimic real videos. However, this approach primarily focuses on keyframes understanding, which is insufficient for scenarios requiring the model to process hundreds of frames at high fps, such as action counting tasks. Besides, for the fine-grained action understanding scenarios, TOPA is unable to capture the fine-grained visual information. For example, in a scene where a person climbs a ladder, it is difficult for TOPA to identify whether the person is going up or down due to the limited capability to capture detailed visual dynamics. Further enhancing TOPA with video instruction tuning might address these limitations which we leave for future work.

# C Broader Impact

**Academic Impact.** TOPA's methodology, which frees the need for costly video-text data collection and large-scale pre-training, lowers the barriers to entry for research and development in video-language understanding technologies. The text-only learning framework of TOPA may inspire researchers with limited resources to engage in cutting-edge multi-modal research, providing a more diverse range of perspectives and contributions to this field.

**Social Impact.** The ultimate objective of TOPA is to develop a general video-language understanding model. Its primary application enables users to extract information from long-form videos without the need for detailed viewing. Moreover, these capabilities for interpreting and managing video content could significantly enhance content moderation systems. Platforms hosting user-generated content could employ sophisticated video-language models to efficiently detect and mitigate the effects of inappropriate or harmful video content.

# D The details of proposed TextVid dataset

We utilize Gemini Pro 1.0 API for our data generation process. We prompt the LLM to create textual videos along with associated annotations. To ensure a diverse dataset that covers a wide of domains, we add condition prompts including different themes, video captions, video events, and the names of main objects. Specifically, we leverage video titles from Howto100M [40], video captions from WebVid2M [3], video events from Ego4D [15], and object from Wordnet [41] as conditions to generate diverse textual videos. For Ego4D condition, we ask the LLM to mimic an ego-centric video to further improve the diversity of the dataset. Table 10 compares vocabulary sizes. Figure 11 shows that Tideos generated under different conditions have different distributions. For each data generation, we prompt the LLM with the task prompt and one of the condition prompts as shown in Figure 13. The statistics TextVid are shown in Table 10. Additionally, we provide Wordcloud of TextVid in Figure 10. The examples of TextVid are shown in Appendix H.

Table 10: Statistics of TextVid.

|  | **TextVid** |
|---|---|
| Generated by | Gemini Pro 1.0 |
| # textual videos | 721K |
| # each condition | Video Title-213k; Video Caption-183K; Video Scenarios-205K; Object-120K |
| # all QA pairs | 3.5M |
| # each question types | What-2.5M; Why-410K; How-287K;Others-254K |
| # all frames | 4.4M |
| Avg. frame | 6.13 |

Table 11: Vocabulary size of Tideos generated under different prompts. We randomly sampled 20,000 global captions from each type of Tideos for comparison.

| | Howto100m | Ego4D | WebVid | WordNet |
|---|---|---|---|---|
| **Vocab Size** | 17492 | 7320 | 15095 | 26486 |

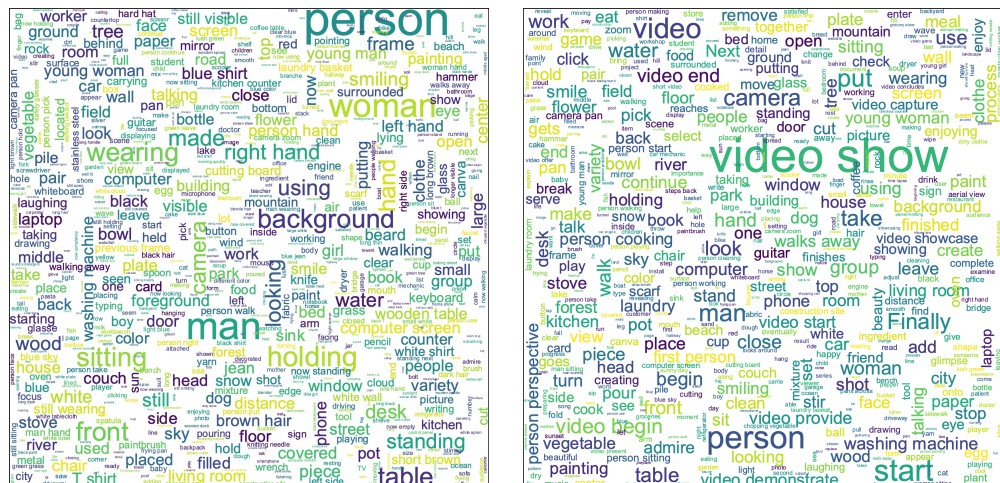

Figure 10: Wordcloud of TextVid. The frame caption (left) and the dense video caption (right).

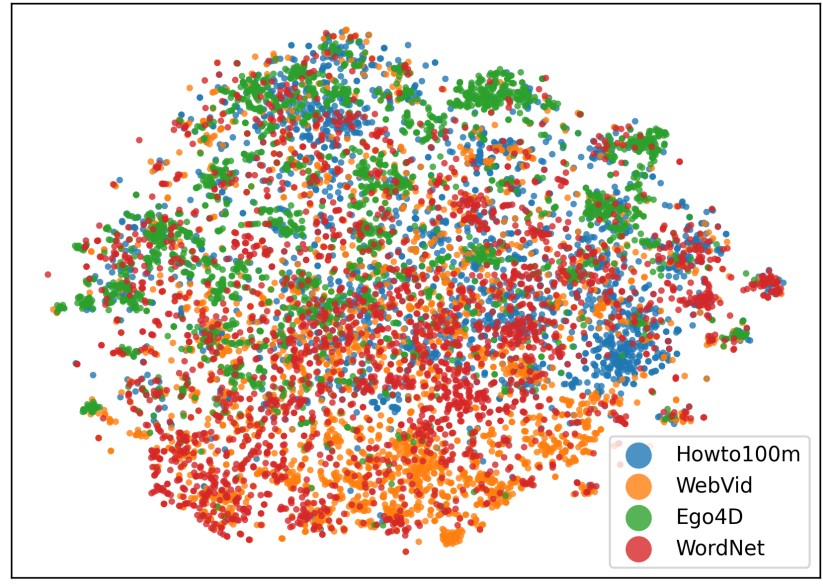

Figure 11: Visualization of Tideo features generated from different type of prompts.

# E Experimental Setup

## E.1 Benchmarks

**EgoSchema** [39] is a challenging long-form video understanding benchmark with 5000 multi-choice questions. The videos in EgoSchema are sourced from Ego-4D [15], with an average length of 3 minutes, distinct from previous benchmarks that focused on shorter, seconds-long videos. The

questions in EgoSchema are manually curated to demand long temporal reasoning. We report results on EgoSchema full set.

**NExT-QA** [72] is a multi-choice video QA benchmark for causal and temporal reasoning, including 5,440 natural videos. The average length of video is 44 seconds. We report results on NExT-QA validation set, which contains 570 videos and 5,000 multiple-choice questions.

**STAR** [70] is a benchmark for situated reasoning. It contains 22K video clips with an average length of 44 seconds. There are 4 different question types in STAR: Interaction (Int.), Sequence (Seq.), Prediction (Pre.), and Feasibility (Fea.). We report results on STAR validation set.

**TVQA** [24] is a benchmark containing 21k video clips with an average length of 76 seconds. We report results on TVQA validation set without subtitles.

**MVbench** [29] is a reorganized benchmark containing 20 video understanding tasks. These tasks are sourced from STAR [70], PAXION [68], MiT Vi [43], FunQA [73], Perception Test [46], Charades-STA [14], MovieNet [18], NTU RGB+D [35], VLN-CE [69] and TVQA [24].

## E.2 The details of training and evaluation.

We leverage Llama2-7B, Llama2-13B [57] and Llama3-8B as the LLM backbone. Additionally, we employ the Llama-adapter [84] with an adaptation embedding length of 50 for efficient finetuning. We utilize CLIP-ViT-L as the multimodal encoder. We employ a simple linear layer to project the CLIP feature into the LLM feature space. The CLIP model and LLM backbone are frozen. The projection layer and additional Llama-adapter are trainable. For text-only pre-alignment, we uniformly sample the Tideos into 10 frames. We train the model on a mixture of tasks comprising Tideo summarization, Tideo QA, multi-choice Tideo QA with the ratio of 1:1:2. TOPA-Llama2-7B and TOPA-Llama3-8B are trained on four 40G-A100 GPUs in one day. TOPA-Llama2-13B is trained in two days. For zero-shot inference, we construct a memory for cross-modal projection, consisting of 2M CLIP text features sampled from the frame captions in the TextVid dataset. We include the training details in Table 12. The actual learning rate is calculated by `base lr ∗ Effective Batchsize`/256.

Table 12: Training hyper-parameters.

| | Model | Training Dataset | Epoch | Effective Batchsize (bs, #GPUs, grad accu) | base lr | Optimizer |
|---|---|---|---|---|---|---|
| Pre-training | TOPA-LLama2-7B | TextVid 721K | 20 | 18x4x4 | 5e-3 | AdamW |
| | TOPA-LLama2-13B | | | 4x4x8 | 8e-3 | weight decay 0.1 |
| | TOPA-LLama3-8B | | | 14x4x8 | 5e-3 | warm up 1 epoch |
| Fine-tuning | TOPA-LLama2-7B | NextQA STAR TVQA | 5 | 20x4x4 | 5e-3 1e-2 5e-3 | |
| | TOPA-LLama2-13B | NextQA STAR TVQA | 5 | 6x4x16 | 2e-3 2e-3 5e-3 | AdamW [36] weight decay 0.1 warm up 1 epochs |
| | TOPA-LLama3-8B | NextQA STAR TVQA | 5 | 20x4x4 | 1e-2 1e-2 5e-3 | |
| Baseline | LLama2-7B | NextQA STAR TVQA | 10 | 20x4x4 | 1e-2 2e-2 2e-2 | |
| | LLama2-13B | NextQA STAR TVQA | 10 | 6x4x16 | 1e-2 2e-2 2e-2 | AdamW weight decay 0.1 warm up 2 epochs |
| | LLama3-8B | NextQA STAR TVQA | 10 | 20x4x4 | 2e-2 2e-2 1e-2 | |

# F Prompts

## F.1 Text-only Training Prompts.

We use the following prompts for Text-only training. The prompts are partially based on [23].

Tideo Multi-choice QA:
Instruction: Choose the correct answer based on the video and question.
Video: $\{\mathbf{f}_1^t,...,\mathbf{f}_n^t\}$.
Question: {Question}.
Choices:
(A): {Option A}. (B): {Option B}. (C): {Option C}. (D): {Option D}. (E): {Option E}.
Answer: The correct choice is {Correct Choice}.

Tideo QA:
Instruction: Predict the answer based on the video and question.
Video: $\{\mathbf{f}_1^t,...,\mathbf{f}_n^t\}$.
Question: {Question}.
Answer: {Answer}.

Tideo Description:
Instruction: Generate a dense description for the video.
Video: $\{\mathbf{f}_1^t,...,\mathbf{f}_n^t\}$.
Description: {Tideo Description}.

## F.2 Prompt for Gemini Blind Evaluation

In Table 6, we use the following prompt for the blind evaluation of Gemini-Pro-1.0 on EgoSchema. The prompt is based on [4].

You are a helpful assistant, an expert in answering questions about videos. You will be given a question about a video and five possible answer options. You will not be provided frames from the video, but still do your best to guess an answer. You are very capable, think step-by-step when answering the question.

Question: <question>

Possible answer choices:
(A): <Choice 1>
(B): <Choice 2>
(C): <Choice 3>
(D): <Choice 4>
(E): <Choice 5>

The response should be in JSON format:
{"Answer": "(X)"}

Figure 12: The multi-choice QA prompts used for the blind evaluation of Gemini-1.0-Pro.

## F.3 Prompts for Data Generation

The prompts for TextVid generation are shown in Figure 13.

## Task Prompt

Please assist in creating a textual dataset that simulates a video with 5-15 frame-level descriptions. The output should be structured in JSON format.

For each frame, you should describe it as following:
**Frame Caption:** Start with a 'Frame Caption' for each frame, providing a comprehensive overview.
**Detailed Captions:** Follow the frame caption with 1-3 detailed captions focusing on specific objects within the frame. Each detailed caption should highlight the main object at the beginning.

After describing the frames, provide:
**Global Video Description:** Provide a summary that synthesizes information from all described frames, offering an overarching narrative of the video.
**QA Pairs:** A set of question-answer pairs related to the video. Questions can vary, including "when," "why," "what," "how," "does/do," "is/are," and "where." Emphasis should be on visual reasoning. Each question will come with five options, including one correct answer. Questions that require integration of multiple frames to deduce the answer are preferred.

Output should structured as following:
```
{
 "Video_Title": " ",
 "Frames": [
  {
   "Frame Caption": " ",
   "Object Name": " ",
   "Object Name": " ",
   "Object Name": " "
  },
  ...
 ],
 "Global_Video_Description": " ",
 "QA Pairs": [
  {
   "Question": " ",
   "Options": {
    "A": " ",
    "B": " ",
    "C": " ",
    "D": " ",
    "E": " "
   },
   "Answer": "D"
  },
  ...
 ]
}
```

## Condition Prompts

**Video title (Howto100M):**
Now, please generate a textual video about {title}.

**Video caption (WebVid):**
Now, please generate a textual video about {caption}.

**Video scenarios (Ego4d):**
Now, please generate an ego-centric video about {scenarios}. This video is taken by {name}.

**Object (Wordnet):**
Now, please generate a textual video about {object name}, {object description}.

Figure 13: The prompts used in TextVid generation.

# G License

The code, model and proposed dataset will be publicly accessible. We use standard licenses from the community. We include the following licenses for the codes, datasets and models we used in this paper.

1. **Benchmarks**

   NExT-QA [72]: MIT

   STAR [70]: Apache

   TVQA [24]: MIT

   EgoSchema [39]: MIT

   MVBench [29]: MIT

2. **Codes**

   LLama-adapter [84] GNU General Public License v3.0

   Flipped-VQA [23]: MIT

3. **Models**

   CLIP [48]: MIT

   LLama2 [57]: Llama 2 Community License Agreement

   LLama3: Meta Llama 3 Community License Agreement

   Gimini-API [56]: Gemini API Additional Terms of Service.

# H    Examples of TextVid

## Textual Video

**Video Title**: Win the Budweiser Shootout in Nascar 07

**Frame Caption 1**: A car race is taking place on a racetrack.
**Racecar:** The racecar is speeding along the track.
**Crowd:** A large crowd is gathered around the race to watch.

**Frame Caption 2**: The racecar is approaching a turn in the track.
**Racecar:** The racecar is preparing to turn the corner.
**Driver:** The driver is focused on the track ahead.
**Stands:** The grandstands are filled with spectators.

**Frame Caption 3**: The racecar is in a fierce battle for the lead.
**Racecar1:** The racecar in front is trying to hold onto its lead
**Racecar2:** The racecar in back is pushing hard to overtake.'
**Track:** The track is lined with advertising billboards.'

**Frame Caption 4**: The racecar is running at the racetrack.
**Racecar:** The racecar is speeding along the straightaway.
**Driver:** The driver is focused on the road ahead.
**Pit crew:** The pit crew is preparing for a pit stop.

**Frame Caption 5**: The racecar is blowing out
**Racecar:** The racecar is damaged and smoking.
**Driver:** The driver is trying to control the car.
**Track:** The track is covered in debris.

**Frame Caption 6**: The driver got out of the car and give up the game.
**Racecar:** The racecar is stopped on the track.'
**Driver:** The driver is walking away from the car.
**Crew:** The crew is running towards the car.

## Annotations

**Detailed Video Descriptions:** The video begins with the racecars speeding around the track, as the driver navigates the turns and battles for the lead. Suddenly, one of the racecars suffers a blowout and crashes, causing the driver to exit the vehicle and surrender the race.

**QA Pairs:**
Question:  What caused the racecar to crash?
Options:
A: A mechanical failure.
B: A collision with another car.
C: A flat tire.
D: A driver error.
E: Bad weather conditions.
Answer: C

Question:  What happened to the driver after the crash?
Options:
A: He was injured and taken to the hospital.
B: He was able to walk away from the car.
C: He was disqualified from the race.
D: He was arrested for reckless driving.
E: He won the race.
Answer: B

Question What is the name of the track where the race is taking place?
Options:
A: Daytona International Speedway.
B: Indianapolis Motor Speedway.
C: Talladega Superspeedway.
D: Charlotte Motor Speedway.
E: Bristol Motor Speedway.
Answer: A

## Textual Video

**Video Title**: Croesus

**Frame Caption 1**: Croesus sits on the throne in grandeur with a golden crown and scepter in hand, exuding wealth and power.
**Croesus:** A wealthy man wearing a golden crown and scepter, sitting on a throne.
**Throne:** A large, ornate chair made of gold and jewels.
**Scepter:** A long, thin staff made of gold, topped with a decorative finial.

**Frame Caption 2**: Croesus holds a scroll with mathematical equations written on it, revealing his intelligence and passion for knowledge.
**Croesus:** A wealthy man wearing a golden crown, holding a scroll
**Scroll:** A long, thin piece of paper with writing on it.
**Equation:** A mathematical expression consisting of variables and operators.

**Frame Caption 3**: Croesus stands before a large group of people, gesturing with his arms as he delivers a speech with confidence and authority.
**Croesus:** A wealthy man wearing a golden crown, standing in front of a crowd.
**Crowd:** A large group of people gathered together.

**Frame Caption 4**: Croesus sits with a wise man, seeking guidance and knowledge from the elder's experience and wisdom.
**Croesus:** A wealthy man wearing a golden crown, sitting with a wise man.
**Wise man:** An elderly man with a long beard, dressed in simple robes.

**Frame Caption 5**: Croesus stands alone in a desolate wasteland, his expression one of despair and regret, as he contemplates the consequences of his actions.
**Croesus:** A wealthy man wearing a golden crown, standing in a wasteland.
**Wasteland :** A barren, empty area of land.

## Annotations

**Detailed Video Descriptions:** This video depicts the life of Croesus, a wealthy and powerful king who was known for his wisdom and intelligence. The video shows Croesus in various settings, including his throne room, his study, and a public square. Throughout the video, Croesus is shown interacting with various people, including scholars, advisors, and commoners. The video ends with Croesus standing alone in a desolate wasteland, contemplating the consequences of his actions.

**QA Pairs:**
Question:  What is Croesus holding in his hand in frame 2?
Options:
A: A book.
B: A scroll.
C: A scepter.
D: A sword.
E: A crown.
Answer: B

Question: Why is Croesus standing in a desolate wasteland in the final frame?
Options:
A: He is lost.
B: He has been defeated in battle.
C: He is contemplating the consequences of his actions.
D: He is seeking solitude.
E: He is hiding from his enemies .
Answer: C

Question: How does Croesus's expression change throughout the video
Options:
A: He becomes increasingly confident .
B: He becomes increasingly worried.
C: He becomes increasingly angry.
D: He becomes increasingly sad.
E: His expression does not change.
Answer: B

## Textual Video

**Video Title**: Make Dough Without Yeast

**Frame Caption 1**: A kitchen counter with flour, salt, baking powder, and olive oil in bowls
**Flour**: A bowl of flour is on the left side of the counter
**Salt**: A bowl of salt is next to the bowl of flour.
**Baking powder**: A bowl of baking powder is next to the bowl of salt.
**Salt**: A bowl of salt is next to the bowl of flour.

**Frame Caption 2**: A woman is adding olive oil to the flour and salt in a bowl
**Woman**: A woman is standing behind the counter, adding olive oil to a bowl of flour and salt.
**Olive oil:** A cup of olive oil is in the woman's left hand.
**Bowl:** A bowl of flour and salt is on the counter in front of the woman.

**Frame Caption 3**: The woman is stirring the ingredients in the bowl.
**Woman:** The woman is standing behind the counter, stirring ingredients in a bowl.
**Spoon:** A spoon is in the woman's right hand
**Bowl:** A bowl of ingredients is on the counter in front of the woman.

**Frame Caption 4**: The woman is adding water to the ingredients in the bowl.
**Woman:** The woman is standing behind the counter, adding water to a bowl of ingredients.
**Water:** A cup of water is in the woman's left hand.
**Bowl:** A bowl of ingredients is on the counter in front of the woman.

**Frame Caption 5**: The woman is stirring the ingredients in the bowl with a fork.
**Woman:** The woman is standing behind the counter, stirring the ingredients in a bowl with a fork.
**Fork:** A fork is in the woman's right hand.
**Bowl:** A bowl of ingredients is on the counter in front of the woman.

**Frame Caption 6**: The woman is kneading the dough on a floured surface.
**Woman:** The woman is standing behind the counter, kneading dough on a floured surface.
**Dough:** A ball of dough is on the counter in front of the woman.
**Surface:** The counter is covered in flour.

**Frame Caption 7**: The woman is shaping the dough into a loaf.
**Woman:** The woman is standing behind the counter, shaping dough into a loaf.
**Dough:** A loaf of dough is on the counter in front of the woman.
**Hands:** The woman's hands are on the dough.

**Frame Caption 7**: The woman is placing the loaf of dough in a baking pan.
**Woman:** The woman is standing behind the counter, placing a loaf of dough in a baking pan.
**Dough:** A loaf of dough is in the woman's hands.
**Pan:** A baking pan is on the counter in front of the woman.
**………..**

## Annotations

**Detailed Video Descriptions:** This video shows how to make dough without yeast. The woman starts by adding olive oil to a bowl of flour and salt. She then stirs the ingredients together and adds water. The woman continues to stir the ingredients until they form a dough. She then kneads the dough on a floured surface and shapes it into a loaf. The woman places the loaf of dough in a baking pan and bakes it in the oven. Once the bread is baked, the woman removes it from the oven and lets it cool.

**QA Pairs:**
Question: What is the woman adding to the flour and salt?
Options:
A: Water.
B: Olive oil
C: Baking powder.
D: Sugar
E: Yeast
Answer: B

Question: What is the woman doing in frame 6?
Options:
A: Stirring the ingredients
B: Kneading the dough
C: Shaping the dough
D: Baking the bread
E: Removing the bread from the oven
Answer: B

Question: Where is the woman placing the loaf of dough in frame 8?
Options:
A: In a bowl
B: On a baking sheet
C: In a baking pan
D: On a cutting board
E: In a plastic bag
Answer: C

Question: Why is the woman stirring the ingredients in frame 3 ?
Options:
A: To combine them
B: To dissolve the salt
C: To activate the yeast
D: To make the dough smooth
E: To prevent the dough from sticking
Answer: A

Question: What is the final product of this video?
Options:
A: Dough
B: Bread
C: Pizza
D: Pasta
E: Cake
Answer: B

## Textual Video

**Video Object Name**: Shouldered Arch

**Frame Caption 1**: A close-up view of a shouldered arch. The arch is made of stone and has a slightly pointed shape. It is supported by two corbels that project into the aperture.
**Shouldered arch:** The shouldered arch is made of stone and has a slightly pointed shape.
**Corbels:** The corbels are made of stone and project into the aperture.
**Lintel:** The lintel is a horizontal stone that spans the opening of the arch.

**Frame Caption 2:** The camera pans out to show the shouldered arch in the context of a larger building. The arch is part of a doorway that leads into a courtyard.
**Shouldered arch:** The shouldered arch is part of a doorway that leads into a courtyard.
**Courtyard:** The courtyard is surrounded by a wall and has a fountain in the center.
**Building:** The building is made of stone and has a tiled roof.

**Frame Caption 3:** A person walks through the doorway into the courtyard. Jose is wearing a long robe and carrying a book.
**Person:** Jose is wearing a long robe and carrying a book.
**Courtyard:** The courtyard is surrounded by a wall and has a fountain in the center.
**Shouldered arch:** The shouldered arch is part of a doorway that leads into a courtyard

**Frame Caption 4:** The camera zooms in on the shouldered arch. The arch is decorated with a carving of a lion's head.
**Shouldered arch:** The shouldered arch is decorated with a carving of a lion's head.
**Lion's head:** An elderly man with a long beard, dressed in simple robes.

**Frame Caption 5:** The camera pans out to show the shouldered arch in the context of a larger building. The arch is part of a window that overlooks the courtyard.
**Shouldered arch:** The shouldered arch is part of a window that overlooks the courtyard.
**courtyard:** The courtyard is surrounded by a wall and has a fountain in the center.

## Annotations

**Detailed Video Descriptions:** This video shows a shouldered arch. The arch is made of stone and has a slightly pointed shape. It is supported by two corbels that project into the aperture. The arch is part of a doorway that leads into a courtyard. The courtyard is surrounded by a wall and has a fountain in the center. The arch is also decorated with a carving of a lion's head.

**QA Pairs:**
Question: What type of arch is shown in the video?
Options:
A: Round arch.
B: Pointed arch.
C: Shouldered arch.
D: Horseshoe arch.
E: Segmental arch.
Answer: C

Question: What is the purpose of the corbels in the arch?
Options:
A: To support the lintel.
B: To decorate the arch
C: To provide ventilation
D: To strengthen the arch
E: To reduce the weight of the arch
Answer: A

Question: What is carved into the stone of the arch?
Options:
A: A lion's head
B: A human face
C: A floral design
D: A geometric pattern
E: Nothing
Answer: A

Question: Where is the arch located?
Options:
A: In a doorway
B: In a window
C: In a wall
D: In a ceiling
E: In a bridge
Answer: A

## Textual Video

**Video Description**: Aerial view of young man by the mountain lake looking at trail map for directions.

**Frame Caption 1**: A man is sitting on a couch, with his hand on his temple
**Man:** A man with a serious expression is sitting on a couch, his head is down and his eyes are closed.
**Sofa:** A beige fabric sofa is at the center of the living room, and a man is sitting on it.
**Hands:** The man's right hand is on his forehead, fingers rubbing his temple.
**Background:** Behind the sofa, a large window with white curtain on the right side.

**Frame Caption 2**: He massages his temple.
**Man:** The man is still sitting on the couch, his expression is still serious and his eyes are still closed.
**Sofa:** The man is sitting on this beige fabric sofa in his living room.
**Hands:** The man's right hand is still on his forehead, applying more pressure.
**Background:** A large window with white curtain is behind the man and the sofa.

**Frame Caption 3**: He seems to be in pain.
**Man:** The man's expression is getting worse, he wrinkles his brows and his mouth is slightly open.
**Sofa:** The man is sitting on a beige fabric sofa in the living room.
**Hands:** The man's right hand is still on his forehead, applying lots of pressure.
**Background:** A large window with white curtain is behind the man and the sofa.

**Frame Caption 4**: He is holding his temple and grimacing.
**Man:** The man's face is scrunched up in pain, his eyes are closed tightly.
**Sofa:** An elderly man with a long beard, dressed in simple robes.
**Hands:** The man's right hand is still on his forehead, applying pressure.
**Background:** A large window with white curtain is behind the man and the sofa.

**Frame Caption 5**: The man seems to be slightly relieved.
**Man:** The man's face looks a little better, he is starting to release the pressure on his forehead.
**Sofa:** The man is sitting on a beige fabric sofa in his living room.
**Hands:** The man's right hand is still on his forehead, but he is not applying pressure anymore.
**Background:** A large window with white curtain is behind the man and the sofa.

## Annotations

**Detailed Video Descriptions:** A man sitting on a couch in pain, rubbing his temple to relieve a migraine headache

**QA Pairs:**
Question: What is the man doing to help relieve his pain?
Options:
A: Rubbing his forehead.
B: Taking medication.
C: Applying a cold compress.
D: Drinking plenty of water.
E: Exercising.
Answer: A

Question: What is the man's expression?
Options:
A: Happy
B: Sad
C: Angry
D: In pain
E: Surprised
Answer: D

Question: What does the man have?
Options:
A: A headache
B: A toothache
C: A stomachache
D: A cold
E: A flu
Answer: A

## Textual Video

**Video Recorder:** Jose

**Frame Caption 1:** A first-person perspective of Jose standing in the kitchen, holding a knife and chopping vegetables on a cutting board.
**Cutting board:** A rectangular wooden cutting board is placed on the kitchen counter.
**Knife:** Jose is holding a sharp, stainless steel knife in his right hand.

**Frame Caption 2:** Jose is cracking an egg into a bowl.
**Man:** The man is still sitting on the couch, his expression is still serious and his eyes are still closed.
**Bowl:** A white ceramic bowl is placed on the kitchen counter.
**Egg:** Jose is holding a brown egg in his left hand.

**Frame Caption 3:** Jose is whisking the egg and other ingredients in the bowl.
**Whisk:** Jose is holding a silver whisk in his right hand.
**Bowl:** The white ceramic bowl is filled with egg and other ingredients

**Frame Caption 4:** Jose is pouring the egg mixture into a pan..
**Pan:** A black nonstick pan is placed on the stovetop.
**Bowl:** Jose is holding the white ceramic bowl in his left hand.
**Spatula:** Jose is holding a black spatula in his right hand.

**Frame Caption 5:** Jose is flipping the omelet in the pan.
**Omelet:** The omelet is cooking in the pan, browning on one side.
**Spatula:** Jose is using the black spatula to flip the omelet.

## Annotations

**Detailed Video Descriptions:** The video shows Jose cooking an omelet. He first chops vegetables on a cutting board, then cracks an egg into a bowl. He whisks the egg and other ingredients in the bowl, then pours the mixture into a pan. He flips the omelet in the pan until it is cooked through.

**QA Pairs:**
Question: What is Jose's primary goal in this video?
Options:
A: To wash dishes.
B: To make an omelet.
C: To clean the kitchen
D: To eat dinner.
E: To chop vegetables.
Answer: B

Question: What type of pan does Jose use to cook the omelet?
Options:
A: A cast iron skillet.
B: A nonstick pan.
C: A Dutch oven
D: A saucepan.
E: A baking dish.
Answer: B

Question: What ingredients does Jose add to the egg mixture?
Options:
A: Salt and pepper.
B: Milk and cheese.
C: Flour and water
D: Meat and vegetables.
E: Sugar and butter
Answer: A

Question: What does Jose do with the omelet after it is cooked?
Options:
A: He serves it with toast.
B: He eats it with a fork.
C: He puts it in a lunch box.
D: He gives it to his dog.
E: He throws it away.
Answer: B

