# OpenReview forum: "TOPA: Extending Large Language Models for Video Understanding via Text-Only Pre-Alignment"
_NeurIPS.cc/2024/Conference — NeurIPS 2024 spotlight_

### Official Review · Reviewer_oFSV · 2024-07-07

**Soundness:** 4
**Presentation:** 4
**Contribution:** 4
**Rating:** 7
**Confidence:** 4

**Summary:**

This paper introduces Text-Only Pre-Alignment (TOPA), a framework designed to extend LLMs for video understanding, without the need for real video data. TOPA leverages textual videos (Tideo), generated by capable LLMs, to mimic real videos. Aided by CLIP's aligned cross-modal space, this  framework despite training on text-only data,  can effectively handle real video input. The effectiveness of TOPA is demonstrated through extensive experiments on video understanding benchmarks.

**Strengths:**

1. This paper propose a novel text-only pre-alignment framework, TOPA, to extend LLM for video understanding. This method leverages LLM-generated text-only data for training, reducing the reliance on video data and human annotations. The text-only learning pipeline introduced in this paper has potential implications for multimodal learning.

2. The introduced TextVid dataset is innovative. The idea of using LLM-generated "Tideo" to mimic real videos and employing CLIP to bridge the Tideo representation with real video representation is particularly interesting

3. The method is evaluated across various video understanding benchmarks, including the challenging EgoSchema and MVbench. The effectiveness of TOPA is demonstrated in multiple settings, including zero-shot evaluation, pretrain-finetune schemes, and data-efficient finetuning. The paper also performs ablation studies to analyze the impact of proposed method.

4. Many different approaches have compared and discussed in this paper, including vidoe-text pretraining approaches, image MLLM-based approaches and LLM-based video agents. This comprehensive comparison could serve as a valuable reference for future research on video understanding.

5. The paper is well written and easy to follow.

**Weaknesses:**

1. The paper could benefit from providing a more detailed comparison between Tideo and real video. For instance, the TextVid dataset likely covers more diverse domains, due to the use of varied prompts during Tideo generation. It could be analyzed in depth.

2. The proposed Tideos are primarily key-frame level representations, which may not fully capture the temporal dynamics of real videos.

3. The experiment results on MVBench should be incorporated into Section 4.1.2 for improved understanding of TOPA. The results on MVBench clearly demonstrate TOPA's strengths and limitations.

**Questions:**

1. The proposed approach regards video as a few key frames. Can TOPA handle scenarios with more frames?

2. Could you provide more details about the finetuning stage? Are both the projector and adapter optimized during finetuning?

3. How scalable is the TOPA framework when applied to larger datasets or more complex video understanding tasks?

**Limitations:**

As stated by the authors, the text-only learning framework is restricted by the imperfectly aligned CLIP model. The projected visual representations are limited in capturing fine-grained visual details.

---

> ### Author Rebuttal · Authors · 2024-08-07
>
> > Weakness1: The paper could benefit from providing a more detailed comparison between Tideo and real video. For instance, the TextVid dataset likely covers more diverse domains, due to the use of varied prompts during Tideo generation. It could be analyzed in depth.
>
> Ans: Thank you for your suggestion. TextVid is notably diverse compared to real video datasets. For instance, previous datasets like Howto100m and Ego4D focus on specific domains, such as instructional videos from YouTube. By using varied condition prompts, we enhance the diversity of Tideos to cover a broader range of domains.
>
> We provide visualizations in the submitted PDF. (a) Figure 1 shows that Tideos generated under different conditions exhibit varying distributions. Tideos-WordNet, in particular, displays the most diverse distribution, scattered across the space. (b) Figure 2 illustrates that Tideos-Howto100m and Tideos-Ego4D focus primarily on human activities, while Tideos-WebVid and Tideos-WordNet cover a broader range of scenarios. (c) Table 3 compares vocabulary sizes, showing that Tideos-WordNet tends to cover more objects.
>
> > Weakness2: The proposed Tideos are primarily key-frame level representations, which may not fully capture the temporal dynamics of real videos.
>
> Ans: Sampling videos into several key frames is a widely adopted approach in vision-language models [1,2,3] due to its efficiency and considerable performance. Understanding videos at high frame rates is indeed challenging and essential for tasks, such as action counting. Recent work has explored this area [4,5]. However, this is not the primary focus of our paper. TOPA may be extended to handle more frames scenarios by incorporating additional local aggregation modules and fine-tuning with a larger number of frames, which we leave it as future work.
>
>
>
> > Weakness3: The experiment results on MVBench should be incorporated into Section 4.1.2 for improved understanding of TOPA. The results on MVBench clearly demonstrate TOPA's strengths and limitations.
>
> Ans: Thank you for your suggestion. Due to page limitations, we could not include the MVBench results in the submitted version. We will try to incorporate the MVBench experiment into the main paper for improved clarity and understanding of TOPA in final version.
>
>
>
> > Q1: The proposed approach regards video as a few key frames. Can TOPA handle scenarios with more frames?
>
> Ans: In this paper, we fix the input frames to 10 for simplicity. There are several approaches to extend TOPA to handle more frames like (a) Frame selection: Choosing a subset of frames based on criteria such as importance or relevance. (b) Additional local aggregation module: Adding a module that aggregates information from local regions to obtain clip-level representation. (c) Fine-tuning with more frames: Finetuning the model with larger number of frames.
>
> > Q2: Could you provide more details about the finetuning stage? Are both the projector and adapter optimized during finetuning?
>
> Ans: We provide detailed finetuing hyper-parameters for each dataset on Table 11. During supervised finetuning, both of the projector and adapter are optimized. We will enhance the presentation of training, fine-tuning, and inference details in the final version of our paper.
>
> > Q3: How scalable is the TOPA framework when applied to larger datasets or more complex video understanding tasks?
>
> Ans: The TOPA framework exhibits considerable potential for scalability across larger datasets and video understanding tasks, due to the innovative data generation method and text-only training. TOPA framework leverages LLMs for data generation, providing several key advantages: (1) the potential size of the dataset, TextVid, is virtually unlimited; (2) the diversity of domains covered by Tideos can be easily expanded by prompting LLMs with specific conditions; (3) the supervision signals can be dynamically generated to meet video understanding task requirements, such as video summarization or video chat.
>
>
>
> Ref:
>
> [1] Yang, Antoine, et al. "Zero-shot video question answering via frozen bidirectional language models."  NeurIPS 2022
>
> [2] Yu, Shoubin, et al. "Self-chained image-language model for video localization and question answering."  NeurIPS 2023
>
> [3] Ko, Dohwan, et al. "Large language models are temporal and causal reasoners for video question answering." EMNLP 2023
>
> [4] Papalampidi, Pinelopi, et al. "A simple recipe for contrastively pre-training video-first encoders beyond 16 frames." CVPR 2024
>
> [5] Balažević, Ivana, et al. "Memory consolidation enables long-context video understanding." ICML 2024

---

> > ### Comment · Reviewer_oFSV · 2024-08-11
> >
> > Thank the authors for the detailed response. Most of my concerns have been addressed. Overall, the proposed text-only pre-alignment framework, including the Tideo dataset and the text-only learning process, is both novel and effective. It offers a new perspective and solution for multimodal alignment, which may inspire future research.

---

> > > ### Author Response · Authors · 2024-08-12
> > >
> > > We're glad that we addressed your concerns, and we appreciate your recognition of our work. We would like to thank you once again for your valuable suggestions, which helped improve our paper. We will ensure that our discussions are reflected in the final version.

---

### Official Review · Reviewer_nFok · 2024-07-12

**Soundness:** 4
**Presentation:** 3
**Contribution:** 4
**Rating:** 8
**Confidence:** 5

**Summary:**

The paper presents Text-Only Pre-Alignment (TOPA), a method to extend LLMs for video understanding without training on real video data. TOPA generates "Textual Videos" using LLMs to simulate real video data, then pre-aligns the LLMs with video modalities using the CLIP model for feature extraction. This approach achieves impressive results across various benchmarks, surpassing previous methods and competing well with recent video understanding models.

**Strengths:**

- The term "Tideo" for textual videos effectively captures the concept.
- The motivation of the work is well-formulated, addressing both the high cost of video training and the challenges of aligning different modalities.
- The authors discuss the modality gap, the misalignment between visual and text modalities, and its impact on performance.
- The related work section is concise and clearly positions the study within the context of previous research.
- The novel idea of creating textual descriptions of videos without having the actual videos, thereby skipping the alignment part between video and text modalities, is interesting. It shows that simulating other modality via text and  tuning LLMs solely on language can be more efficient than combining multiple modalities.
- The proposed method is thoroughly evaluated across a range of tasks and models.

**Weaknesses:**

1. L42: The authors describe subtitles as "frame-level descriptions," but subtitles are actually spoken speech extracted from videos and have intrinsic context and alignment issues [HowTo100M].
2. The description of the training process and stages is confusing. It needs to be clearer what is trained when and how. The subsection "Video-LLM alignment" is confusing given the title "text-only pre-alignment." The presentation would benefit from an overview of high-level steps for training/inference, followed by detailed descriptions. The dataset section lacks references to the appendix for exact prompt details.

**Minor:**

- The references should be clickable.
- Correct “Egoschema” to “EgoSchema.”
- The statements “1. Intrinsic complexity of the video modality” and “2. The limitations of web language supervision” should include references to support these claims.

**Questions:**

**Questions:**

1. Have you considered using combined pretraining with both Tideo and multi-modal data?
2. Does the Video-LLM alignment include temporal aggregation of frames?
3. Could you provide ablation studies on how the prompts and different parts of the dataset affect downstream performance, such as the inclusion of condition prompts, the influence of global descriptions, and various QA pairs? It would also be interesting to see how complex the Tideos are and the consistency of frame descriptions. Could the proposed adaptation of LLMs be trained using only text data from available video datasets?
4. Have you considered creating textual descriptions of frames from the original videos (for evaluation) and comparing zero-shot evaluation text vs. videos?

Main Concern:
The quality of the generated video descriptions (Question 3).

**Limitations:**

The limitations are discussed and cover the main disadvantages of the current model.

---

> ### Author Rebuttal · Authors · 2024-08-07
>
> Overall, thank you for your careful reading, detailed feedback, and valuable suggestions. We will revise our paper based on the discussion.
>
> > Weakness1: L42: The authors describe subtitles as "frame-level descriptions," but subtitles are actually spoken speech extracted from videos and have intrinsic context and alignment issues [HowTo100M].
>
> Ans: We agree that "frame-level descriptions" is not an accurate term in this context. Our intention was to highlight that subtitles primarily provide local context and lack the long-term temporal supervision necessary for understanding longer videos, such as those in EgoSchema. We will revise this section for clarity, address [Minor3], and include a discussion on the challenges of visual-textual misalignment [1] [2].
>
> [1]  Lin, Yijie, et al. "Multi-granularity correspondence learning from long-term noisy videos."  ICLR 2024
>
> [2] Han, Tengda, Weidi Xie, and Andrew Zisserman. "Temporal alignment networks for long-term video." CVPR 2022
>
> > Weakness2:The description of the training process and stages is confusing. The subsection "Video-LLM alignment" is confusing given the title "text-only pre-alignment."
>
> Ans: We appreciate the feedback and will revise the paper for clarity. Here is a brief response.
>
> 1. **Training and Inference**.  At "text-only pre-alignment" stage, the model is trained on the TextVid dataset, where the LLM learns to process continuous CLIP text features.
>
>    Then, the text-only pre-aligned model can be applied for real videos in two ways:
>
>    (a) Zero-shot inference (Sec. 4.1). The visual features of test videos are projected into textual features space, and then processed by LLM.
>
>    (b) Finetuning on downstream video datasets (Sec. 4.2). The model is further fine-tuned and evaluated using video features.
>
> 2. **Video-LLM alignment and text-only pre-alignment.** Video-LLM alignment involves aligning LLMs with video modalities, enabling the LLM to process video inputs. Our text-only pre-alignment is a specialized approach within this framework, leveraging text-only data for training.
>
>
>
> > **Q1**: Have you considered using combined pretraining with both Tideo and multi-modal data?
>
> Ans: We attempted pretraining with both Tideos and the WebVid dataset but did not observe performance improvements. We attribute this to the limited supervision signals from the short video captions in WebVid. We believe the use of Tideos in this paper effectively highlights its features and advantages. Integrating Tideos with multimodal data will be explored in future work.
>
> > **Q2**: Temporal aggregation of frames.
>
> Ans: We haven’t designed extra temporal aggregation modules in this work. We assume the LLM can handle temporal aggregation since frames are represented as a sequence of embeddings within the LLM. Table 6 shows TOPA benefits from using more frames, suggesting that the LLM can achieve temporal aggregation.
>
> > **Q3**: Ablation studies on TextVid ... Consistency of frame descriptions ...Trained using text data from video datasets
>
> Ans:
>
> 1. **Condition Prompts Enhance Tideo Diversity.** We aim to enhance the diversity of Tideos through varied condition prompts. We provide visualizations in the submitted PDF. (a) Figure 1 shows that Tideos generated under different conditions have different distributions, with Tideos-WordNet being the most diverse and widely dispersed. (b) Figure 2 illustrates that Tideos-Howto100m and Tideos-Ego4D focus primarily on human-centric activities, while Tideos-WebVid and Tideos-WordNet cover more scenarios. (c) Table 3 compares vocabulary sizes, revealing that Tideos-WordNet encompasses more objects.
>
>    **Diversity and performance**. Our early experiments found that Tideos-WebVid and Tideos-WordNet do not improve downstream performance. It's because the video understanding benchmarks, such as EgoSchema, primarily focus on human-centric activities, which are more closely aligned with Tideos-Ego4D and Tideos-Howto100m. Consequently, Tideos-WebVid and Tideos-WordNet offer limited additional benefits. We look forward to future open-domain video understanding benchmarks to better assess the impact of the our diverse Tideos.
>
> 2. **Global descriptions and QA pairs.** In submitted PDF, an ablation study (Table 1) shows that multi-choice Tideo-QA and Tideo Summarization tasks enhance the performance. For further results and detailed analysis of the multi-choice QA tasks, please see Appendix A.2.
>
> 3. **Tideos Consistency:** In Appendix H, we provide examples of Tideos. The frame descriptions, including frame captions and object captions, show strong consistency.
>
> 4. **Text data from available video datasets may not be suitable for text-only pre-alignment.**  Our text-only pre-alignment relies on Tideos and corresponding supervision. Text data from video datasets, such as Howto100M, provide insufficient supervision for effective training. Additionally, the textual data associated with these videos, such as narrations, is often less detailed compared to our Tideos.
>
> > **Q4:** Have you considered creating textual descriptions of frames from the original videos (for evaluation) and comparing zero-shot evaluation text vs. videos?
>
> Thank you for your insightful suggestion. We add an experiment to study this. We used LaViLa-xl [1] as the video captioner. We divided each test video into 10 clips, each containing 4 frames, and generated clip-level captions for them. These clip-level captions are then processed by the CLIP text encoder to produce 10 textual features, which serve as input for the LLMs.
>
> Results in Table 2 reveal that both project-based and caption-based inference approaches help to mitigate the domain gap issue. Using textual features derived from video captions achieves better results, benefiting from clip-level captions. This demonstrates the flexibility of TOPA and its applicability to various scenarios.
>
> Ref: [1] Zhao, Yue, et al. "Learning video representations from large language models." CVPR 2023

---

> > ### Comment · Reviewer_nFok · 2024-08-12
> >
> > I have carefully reviewed the authors' responses as well as the other reviews. I appreciate the authors' thorough comments, which have addressed all of my concerns. I'm looking forward to the revised version of the paper. I am happy with the current evaluation and discussions.

---

> > > ### Author Response · Authors · 2024-08-12
> > >
> > > We're glad that we addressed your concerns, and we appreciate your recognition of our work. We would like to thank you once again for your valuable suggestions, which helped improve our paper. We will ensure that our discussions are reflected in the final version.

---

### Official Review · Reviewer_kuEy · 2024-07-13

**Soundness:** 2
**Presentation:** 2
**Contribution:** 2
**Rating:** 4
**Confidence:** 5

**Summary:**

The authors introduce Text-Only Pre-Alignment (TOPA), a novel approach that extends large language models (LLMs) for video understanding without pre-training on real video data. TOPA generates Textual Videos, comprising continuous textual frames and annotations to simulate video-text data, and uses these to pre-align a language-only LLM with the video modality. The CLIP model aligns image and text modalities, bridging the gap between textual and real videos. TOPA encodes continuous textual frames as CLIP text features, analogous to CLIP image features, thus aligning the LLM with real video representations.

**Strengths:**

The authors present a novel strategy for approaching video understanding.

The proposed models perform well on Egoschema, NextQA, STAR, and TVQA using the Llama2 and Llama3 LLMs.

**Weaknesses:**

In Table 1, the author did not include key papers such as LifelongMemory[1], Video-Agent: A Memory-Augmented Multimodal Agent for Video Understanding[2], LangRepo[3], and MVU[4]. LifelongMemory achieves 68% and 62.1% on the subset and the full set of EgoSchema, respectively, and should be included in the table to allow readers and reviewers to understand the relative performance, considering the size of the LLM. Additionally, LangRepo and MVU use open-source LLMs (Mixtral7B, Mixtral-8×7B, Llama-2-7b-Chat, Gemma-7b-IT, and Mistral-7B-Instruct) and should also be included to provide a comprehensive comparison. Furthermore, the table is missing the performance on the subset, whereas other papers report performance on both the subset and the full set. Including this information is crucial to observe how accuracy changes from the subset to the full set. Similarly, in Table 2, the author did not include key papers such as LangRepo, MVU, VideoAgent, and LLoVi, making it difficult for readers and reviewers to compare the proposed models with existing works.

https://arxiv.org/abs/2312.05269v1
[1] Ying Wang, Yanlai Yang, and Mengye Ren. Lifelongmemory: Leveraging llms for answering queries in long-form egocentric videos, 2023.

https://arxiv.org/abs/2403.11481
[2] Yue Fan, Xiaojian Ma, Rujie Wu, Yuntao Du, Jiaqi Li, Zhi Gao, and Qing Li. Video-agent: A memory-augmented multimodal agent for video understanding, 2024

https://arxiv.org/abs/2403.14622
[3] Kumara Kahatapitiya, Kanchana Ranasinghe, Jongwoo Park, and Michael S Ryoo. Language repository for long video understanding, 2024.

https://arxiv.org/abs/2403.16998v1
[4] Kanchana Ranasinghe, Xiang Li, Kumara Kahatapitiya, Michael S. Ryoo. Understanding Long Videos in One Multimodal Language Model Pass, 2024

**Questions:**

1. Can you add the performance of the proposed models on the subset of EgoSchema?
2. Can you record the inference time of your proposed method on the entire dataset and compare it with the inference times of existing works, including those listed in this review? There is a concern that the visual-to-text feature conversion in zero-shot mode may slow down the entire model.
3. Can you attempt to use the largest open-source models and show the zero-shot performance on the EgoSchema dataset? It is important to assess whether the proposed model can scale up to larger LLMs and if its performance can be comparable to existing works using closed-source LLMs.

I will reevaluate the proposed model based on the answers to the questions.

**Limitations:**

I mentioned some concerns about the work in the weakness section.

---

> ### Author Rebuttal · Authors · 2024-08-07
>
> > **Weaknesses:** In Table 1, the author did not include key papers such as LifelongMemory[1], Video-Agent: A Memory-Augmented Multimodal Agent for Video Understanding[2], LangRepo[3], and MVU[4]. LifelongMemory achieves 68% and 62.1% on the subset and the full set of EgoSchema, respectively, and should be included in the table to allow readers and reviewers to understand the relative performance, considering the size of the LLM. Additionally, LangRepo and MVU use open-source LLMs (Mixtral7B, Mixtral-8×7B, Llama-2-7b-Chat, Gemma-7b-IT, and Mistral-7B-Instruct) and should also be included to provide a comprehensive comparison. Furthermore, the table is missing the performance on the subset, whereas other papers report performance on both the subset and the full set. Including this information is crucial to observe how accuracy changes from the subset to the full set. Similarly, in Table 2, the author did not include key papers such as LangRepo, MVU, VideoAgent, and LLoVi, making it difficult for readers and reviewers to compare the proposed models with existing works.
>
> Ans: Thank you for your valuable suggestions and for highlighting these related papers. We will include and discuss them in the final version of our paper for a more comprehensive comparison. We kindly note that these papers are appeared/updated on arXiv in **late March 2024** and should be considered as **contemporaneous work**.
>
> **Discussion on LifelongMemory, Video-Agent, LangRepo, and MVU.** We have categoried existing video understanding approaches into 4 categories in Section 4, including *Web video pre-training approaches*, *Adapt image MLLMs for video understanding*, *LLM-based video agents*, and our *Text-only Pre-alignment*. We should note that our *Text-only Pre-alignment* different from previous work that levearges text-only data for video-LLM alignment. We'd like to categorize **LifelongMemory, Video-Agent and LangRepo** into *LLM-based video agents* approaches that leverage VLMs tools convert visual information into textual, and then process video understanding task via LLMs. **MVU** belongs to *Adapt image MLLMs for video understanding*, focusing on adapting image-MLLMs for video understanding. We will update the tables to include results from these papers for a more comprehensive comparison.
>
> **Performance on EgoSchema subset & Evaluation approaches for multi-choices QA**. We report results for the EgoSchema subset in Appendix A.2. We analyze the performance gap between the subset and the full set and discuss the impact of various evaluation methods, including similarity-based, logits-based, and LLM-selection approaches. Appendix A.2 reveals that logits-based and similarity-based approaches perform well on the EgoSchema subset. However, these approaches show limited effectiveness on the full set, resulting in a significant performance gap. This gap may be due to the differing linguistic structures of the two sets: the subset features more similar sentence structures with slight variations, making it easier to distinguish using these methods. In contrast, the full set contains a wider variety of choices, which poses challenges for logits and similarity-based approaches.
>
> We notice that LangRepo and MVU use log-likelihood-based selection for EgoSchema evaluation, which aligns with our discussion in Appendix A.2. We'll include a further discussion about these two contemporaneous work on this section.
>
> >Question1: Can you add the performance of the proposed models on the subset of EgoSchema?
>
> Answered in Weakness.
>
> >  Question 2: Can you record the inference time of your proposed method on the entire dataset and compare it with the inference times of existing works, including those listed in this review? There is a concern that the visual-to-text feature conversion in zero-shot mode may slow down the entire model.
>
> Ans: Thank you for your suggestion. We provide inference time on the following table. We conduct the experiment on a single A100 GPU and report the inference time per sample. The main computational cost of TOPA comes from LLM inference, while the conversion of visual-to-text features adds negligible overhead. Notably, TOPA is more efficient than video-agent approaches, which require multiple additional calls of VLMs.
>
> | Method                      | Core LLM/VLM   | Visual Encoding | Visual-to-Text | LLM inference | All    |
> | --------------------------- | -------------- | --------------- | -------------- | ------------- | ------ |
> | TOPA-LLama2-13B             | LLama2-13B     | 0.0241          | 0.0003         | 0.4998        | 0.5242 |
> | SF-VLM [4]                  | LLaVA-v1.5-13B | -               | -              | -             | 0.9794 |
> | LLoVi (results from MVU[4]) | -              | -               | -              | -             | 207    |
>
>
>
> > Question 3: Can you attempt to use the largest open-source models and show the zero-shot performance on the EgoSchema dataset?
>
> We provide the results of TOPA using Llama2-7B, Llama2-13B, and Llama3-8B across various experiments, showing that TOPA benefits from a more powerful and larger-scale LLM backbone. Exploring the performance of TOPA with even more capable models, such as Llama3.1-70B, would be valuable. However, the training overhead for such large models is currently prohibitive for our research.

---

> ### Author Response · Authors · 2024-08-12
> **Looking Forward to Further Discussion**
>
> Dear Reviewer,
>
> Thank you once again for your valuable suggestions. We hope that our response has adequately addressed your concerns. If you have any further questions or comments, please do not hesitate to reach out. We look forward to any further discussion.
>
> We would like to bring to your attention that the discussion period is scheduled to close on August 13 at 11:59 PM AoE.
>
> Thank you for your time and consideration.
>
> Best regards,
>
> The Authors

---

> > ### Comment · Area_Chair_HtLU · 2024-08-12
> >
> > Dear Reviewer kuEy,
> >
> > The reviews of this paper are diverging and thus your further input is needed. Can you please carefully read the other reviews and the responses from the authors and share any updated thoughts?
> >
> > Do you still think the paper needs improvements to be accepted even after reading the other two reviews that are strongly supporting the paper?
> >
> > AC

---

> > ### Comment · Reviewer_kuEy · 2024-08-13
> > **Benchmark Comparison and Training Datasets**
> >
> > LangRepo as archived on March 21, 2024, which is two months earlier than the submission deadline, so it should have been included in the authors' Table 1  based on the NeurIPS submission guidelines.
> >
> > Additionally, the authors claim that their model outperforms VideoAgnet in lines 213-214 or sentence was miswritten. TOPA is not shown to outperform VideoAgent in Table 1 and table lacks existing models with the similar model size, making it difficult to assess where TOPA actually stands in the context of very long-form video. At a minimum, the authors should have compared performance by either applying Llama2-13B to VideoAgent or applying GPT-4 to their model and explaining the performance differences. Outperforming LLoVi is not sufficient. Furthermore, since TOPA was trained on the Ego4D datase, which includes the EgoSchema videos, the authors cannot claim that the TOPA accuracy in Table 1 is a zero-shot result.
> >
> > The authors also need to address why the accuracy improves when evaluating from the EgoSchema subset to the fullest, while all existing models have shown a performance drop, a pattern generally observed in other datasets. This raises the question of whether this improvement is because TOPA was trained on Ego4D and it is not a zero-shot performance.
> >
> > Based on this reason, I lowered my rating to reject

---

> ### Author Response · Authors · 2024-08-13
> **Clarification1: Our approach——Text-only Pre-alignment**
>
> > LangRepo as archived on March 21, 2024, which is two months earlier than the submission deadline, so it should have been included in the authors' Table 1 based on the NeurIPS submission guidelines.
>
> >Additionally, the authors claim that their model outperforms VideoAgnet in lines 213-214 or sentence was miswritten. TOPA is not shown to outperform VideoAgent in Table 1 and table lacks existing models with the similar model size, making it difficult to assess where TOPA actually stands in the context of very long-form video. At a minimum, the authors should have compared performance by either applying Llama2-13B to VideoAgent or applying GPT-4 to their model and explaining the performance differences. Outperforming LLoVi is not sufficient. Furthermore, since TOPA was trained on the Ego4D datase, which includes the EgoSchema videos, the authors cannot claim that the TOPA accuracy in Table 1 is a zero-shot result.
>
> > The authors also need to address why the accuracy improves when evaluating from the EgoSchema subset to the fullest, while all existing models have shown a performance drop, a pattern generally observed in other datasets. This raises the question of whether this improvement is because TOPA was trained on Ego4D and it is not a zero-shot performance.
>
> > Based on this reason, I lowered my rating to reject
>
> -----
>
> ###  Thank you for the detailed feedback. However, we believe there are many misunderstandings regarding our paper.
>
> # Our approach——Text-only Pre-alignment:
>
> ### **(1) TOPA is NOT a video agent approach.**
>
> While both TOPA and video agents [1,2,3,4] utilize LLMs for video understanding, they are based on fundamentally different core concepts and technical frameworks, representing distinct lines of research.
>
> In this paper, we propose text-only pre-alignment to extend LLMs for video understanding. We propose *Tideo*, consisting of sequential textual frames, to mimic real video. During text-only pre-alignment, the LLM learns to process *Tideo*, which is represented as **sequential CLIP textual features**. During zero-shot inference, real videos are first representated as sequential CLIP visual features. These **visual features** are then projected into the CLIP textual space and processed by the pre-aligned LLM.
>
> Video agents typically employ an LLM as a central agent to iteratively identify and compile key information to answer questions, using VLMs as tools to convert video content into **textual descriptions**.
>
> The major and essential difference is that TOPA takes **video features** as LLM's input, while video agents take **textual descriptions** as LLM's input.
>
> ### **(2) For zero-shot evaluation, TOPA doesn't train on any videos, let along Ego4D videos.**
>
> >  Furthermore, since TOPA was trained on the Ego4D datase, which includes the EgoSchema videos, the authors cannot claim that the TOPA accuracy in Table 1 is a zero-shot result.
>
> > This raises the question of whether this improvement is because TOPA was trained on Ego4D and it is not a zero-shot performance.
>
> For zero-shot evaluation, the TOPA model is trained on generated **text-only** TextVid dataset. There are **no videos** involved in our text-only pre-alignment process.
>
> During the data generation process, we use four types of prompts for diverse Tideo generation, one of which is based on "scenarios" from Ego4D metadata. These "scenarios" are very high-level descriptions like: ```"Watching tv","Cleaning / laundry","Potting plants (indoor),","Scooter mechanic","Eating","Walking on street","Working out outside","Playing with pets"```. We do not believe that the use of such general scenarios compromises the zero-shot setting.
>
> Moreover, most video agents, such as VideoAgent [1], LLoVi [3], and LangRepo [4] use LaViLa[7] as the video captioner, which is **pre-trained on video-narration pairs from Ego4D**. This involves far more detailed information than the high-level scenarios we employ.

---

> ### Author Response · Authors · 2024-08-13
> **Clarification2: Comparison with video agents approaches**
>
> # Comparison with video agents approaches
>
> In this paper, we discussed various video understanding approaches, including video agents. Our goal is to present diverse perspectives on video understanding, rather than to assert that TOPA is the ultimate solution.
>
> ### **(3) The final version will include all the related works we discussed.**
>
> > LangRepo as archived on March 21, 2024, which is two months earlier than the submission deadline, so it should have been included in the authors' Table 1 based on the NeurIPS submission guidelines.
>
> We are glad to include and discuss more related works for comprehensive understanding in our final version.
>
> However, it is important to emphasize once again that **TOPA is not a video agent**. The comparion of these new video agents doesn't affect the main contributions of this paper. Besides,  We have already discussed and compared several representative video agents [1,2,3].
>
> ### **(4) We don't claim that "TOPA outperforms VideoAgent [1]"**
>
> > Additionally, the authors claim that their model outperforms VideoAgnet in lines 213-214 or sentence was miswritten.
>
> Line 213-214 write: *"TOPA outperforms previous image-based adaptation approach IG-VLM and video agents LLoVi and Vamos with the same scale LLM (Llama2-7B and Llama2-13B)"*. Here, "video agents" specifically refer to LLoVi and Vamos, not VideoAgent [1].
>
> ### **(5) Comparison with VideoAgent [1].**
>
> > At a minimum, the authors should have compared performance by either applying Llama2-13B to VideoAgent or applying GPT-4 to their model.
>
> TOPA need train on TextVid dataset, which involves LLM backforward, making it challenging for us to use GPT-4 as the LLM backbone.
> In the table below, we observed a significant performance drop in VideoAgent when using Llama2-70B as the LLM backbone. Notably, TOPA with Llama2-13B outperforms VideoAgent with Llama2-70B, highlighting TOPA's effectiveness.
>
>
> |                | LLM            | EgoSchema Subset |
> | :------------- | :------------- | :--------------: |
> | VideoAgent [1] | **GPT-4**      |       60.2       |
> | VideoAgent [1] | **LLama2-70B** |       45.4       |
> | TOPA           | **LLama2-13B** |       51.2       |

---

> ### Author Response · Authors · 2024-08-13
> **Clarification3: Performance on EgoSchema fullset and subset**
>
> # Performance on EgoSchema fullset and subset.
>
> > The authors also need to address why the accuracy improves when evaluating from the EgoSchema subset to the fullest, while all existing models have shown a performance drop, a pattern generally observed in other datasets. This raises the question of whether this improvement is because TOPA was trained on Ego4D and it is not a zero-shot performance.
>
> ### **(6) EgoSchema subset (500) is a part of EgoShema fullset (5031).**
>
> EgoSchema is focused entirely on evaluation: the hidden full set (test set) consists of 5031 videos via an evaluation server, of which 500 were publicly released for validation. **Therefore, we believe that reporting and comparing results on the full set is valid and convinced, as presented in the EgoSchema paper [8].**
>
> ### **(7) We don't leverage Ego4D videos or data for training.**
>
> TOPA don't train on Ego4D videos or narrations. We only use "scenarios" from Ego4D meta data like ```"Watching tv","Cleaning / laundry","Potting plants (indoor),","Scooter mechanic","Eating","Walking on street","Working out outside","Playing with pets"``` as prompts for Tideo generation.
>
> In contrast, most video agents, such as VideoAgent [1], LLoVi [3], and LangRepo [4] use LaViLa[7] as the video captioner, which is **pre-trained on video-narration pairs from Ego4D**. MC-ViT-L [6] finetuned their model on Ego4D.
>
> ### **(8) The performance gap largely related to evaluation approach.**
>
> As shown in Appendix A.2, we found that most approaches, which evaluate with logits and video-text similarity, share a huge performance gap between subset and full set. **TOPA, when evaluated using the logits method, also experiences this common performance drop.** While for LLM selection approaches, like VideoAgent [1] and TOPA-LLama2-13B, exhibit a smaller gap or no gap. In Appendix A.2, we suggest that this discrepancy may be due to differences in the linguistic structure of the choices.
>
> **Why does TOPA-LLama2-13B show no performance gap?** TOPA uses different training strategies, different inference strategies, different training data and different LLM backbone, compared to other approaches. It's challenging to figure out "why our model performs consistently". Besides, the performance gap phenomenon and evaluation approach is not the focus of this paper. Instead, it would be more appropriate for other approaches, especially those focus on evaluation methods, to investigate why they underperform on the EgoSchema full set.
>
>
>
>
> |                                        | Eval mode         | EgoSchema Subset (500) | EgoSchema Fullset (5031) |  Gap  |
> | -------------------------------------- | ----------------- | :--------------------: | :----------------------: | :---: |
> | LongViViT [5]                          | **Similarity**    |          56.8          |           33.3           | -23.5 |
> | MC-ViT-L [6]                           | **Similarity**    |          62.6          |           44.0           | -18.6 |
> | LangRepo-Mixtral-8×7B-(12B active) [4] | **LLM logits**    |          66.2          |           41.2           | -25.0 |
> | **TOPA-LLama2-13B**                    | **LLM logits**    |          67.5          |           41.6           | -25.9 |
> | VideoAgent (GPT-4) [1]                 | **LLM selection** |          60.2          |           54.1           | -6.1  |
> | **TOPA-LLama2-13B**                    | **LLM selection** |          51.2          |           51.0           | -0.2  |
>
> Ref:
>
> [1] **(ECCV 2024)** Wang, Xiaohan, et al. "Videoagent: Long-form video understanding with large language model as agent."
>
> [2] **(CVPR 2024)** Min, Juhong, et al. "MoReVQA: Exploring Modular Reasoning Models for Video Question Answering."
>
> [3] **(arXiv 2023.12)** Zhang, Ce, et al. "LLoVi: A simple llm framework for long-range video question-answering."
>
> [4] **(arXiv 2024.3)** Kumara Kahatapitiya, Kanchana Ranasinghe, Jongwoo Park, and Michael S Ryoo. Language repository for long video understanding, 2024.
>
> [5] **(CVPR 2024)** Papalampidi, Pinelopi, et al. "A simple recipe for contrastively pre-training video-first encoders beyond 16 frames."
>
> [6] **(ICML 2024)** Balažević, Ivana, et al. "Memory consolidation enables long-context video understanding."
>
> [7] **(CVPR 2023)** Zhao, Yue, et al. "Learning video representations from large language models."
>
> [8] **(NeurIPS)** Mangalam, Karttikeya, Raiymbek Akshulakov, and Jitendra Malik. "Egoschema: A diagnostic benchmark for very long-form video language understanding."

---

> > ### Comment · Reviewer_kuEy · 2024-08-14
> >
> > Thank you once again for the detailed response. The captioners used by VideoAgent and LLoVi were pre-trained on Ego4D while excluding the EgoSchema videos. However, I remain concerned that TOPA was trained using "scenarios" from the Ego4D metadata, which includes EgoSchema scenarios. It would be clearer to claim zero-shot if the authors could demonstrate the performance of TOPA when trained on Ego4D excluding the EgoSchema scenarios.
> >
> > I will raise the score to weak reject, as the authors have addressed some concerns regarding performance against VideoAgent.

---

> ### Author Response · Authors · 2024-08-14
> **Regarding the use of Ego4D "Scenarios"**
>
> Thank you for your feedback.
>
> We are currently conducting an ablation study on the use of "scenarios" prompts from Ego4D. The current results are as follows:
>
> |                 | Epoch | Training data                                 | EgoScheme SubSet (500) | EgoSchema Fullset (5031) |
> | --------------- | :---: | :-------------------------------------------- | :--------------: | :---------------: |
> | TOPA-LLama2-13B |  18   | All Tideos (721K)                             |       51.2       |       51.0        |
> | TOPA-LLama2-13B |   8   | Tideos excluding the Ego4D "scenairos" (516K) |       48.4       |       51.3        |
>
> These results indicate that Tideos-Ego4D does not significantly impact performance on the full EgoSchema dataset, although there is some effect on the subset. We believe this outcome demonstrates that the use of high-level "scenarios" does not lead to special information leakage within the EgoSchema benchmark.
>
> In fact, many of the "scenarios" in the Ego4D metadata represent common human activities. Considering the scale and diversity of Tideos, we believe that "scenarios" in Ego4D are well-covered by the other three types of Tideos, as shown in the PDF. Thus, excluding Tideo-Ego4D does not substantially affect performance on EgoSchema.
>
> Furthermore, we disagree with the notion that our use of high-level "scenarios" is not a zero-shot approach, while the use of same-domain videos and narrations (used in captioner of LLoVi and VideoAgent) is considered a zero-shot approach. From any perspective, the information contained in high-level, common "scenarios" is far less than that provided by videos and narrations from the same dataset.
>
> In our final version, we will clearly state our position on the zero-shot setting and specify the exact data we used.
>
> ## **About "Scenarios" in Ego4D meta:**
>
> Some "Scenarios" in Ego4D meta (Without seletion): ```"Watching tv","Cleaning / laundry","Potting plants (indoor),","Scooter mechanic","Eating","Walking on street","Working out outside","Playing with pets","Bike mechanic","Doing yardwork / shoveling snow","Eating","Farmer","Baker","Cleaning / laundry","Crafting/knitting/sewing/drawing/painting","jobs related to construction/renovation company","Carpenter"```
>
> We use these common scenarios as prompts to generate Tideo, and we don't believe these common scenarios  would involve the leakage of specific video information.
>
> Additionally, most of "Scenarios" are repeated multiple times in Ego4D meta.
>
> For examples, ```Carpenter: 332, Eating:497, Cleaning:1068, Playing with pets:105, Watching tv:154```
>
> It means that even if we exclude all the video information included in EgoSchema, we can still obtain a similar 'Scenarios' corpus for Tideo generation.

---

### Author Rebuttal · Authors · 2024-08-07

# General Response:

We greatly appreciate the reviewers' careful reading and insightful feedback on our work. It's encouraging to receive comments recognizing our work's **novel idea, good motivation, thorough experiments and promising results**. We have respond the concerns raised and are open to further discussions.

We received many valuable suggestions to further improve our paper, and we'd like to share some revision plans here.

(1) **Including and discussing related papers (Reviewer kuEy).** Video understanding is a rapidly evolving field, with numerous new works emerging, such as video-agent approaches. Although we have tried to make comprehensive comparisons with various methods in our submitted version, Reviewer kuEy pointed out some contemporaneous work that we overlooked. We appreciate this feedback and will further discuss and incorporate these references to provide a more comprehensive comparison.

(2) **Providing more analysis on Tideos (Reviewers nFok and oFSV).** We'll include the visualizations to illustrate the diversity of Tideos and the impact of the prompts used in Tideo generation.

(3) **Improving the presentation on pretraining/finetuning/inference (Reviewers nFok and oFSV).**  We will provide a clearer overview of the high-level steps, including text-only pre-alignment, finetuning, and various zero-shot inference strategies, and offer more details.





# Brief Introduction of the PDF:

(1) **Table 1 (suggested by Reviewer nFok):** An ablation study shows that multi-choice Tideo-QA and Tideo Summarization tasks enhance the performance. For further results and detailed analysis of the multi-choice QA tasks, please see Appendix A.2.

(2) **Table 2 (suggested by Reviewer nFok):** We add an experiment to study inference with textual features of videos. We used LaViLa-xl [1] as the video captioner. Results in Table 2 reveal that both project-based and caption-based inference approaches help to mitigate the domain gap issue. Using textual features derived from video captions achieves better results, benefiting from clip-level captions. This demonstrates the flexibility of TOPA and its applicability to various scenarios.

(3) **Figure 1, Figure 2, and Table 3 (suggested by Reviewer nFok and oFSV):** Visualizations of diversity of Tideos. (a) Figure 1 shows that Tideos generated under different conditions have different distributions, with Tideos-WordNet being the most diverse and widely dispersed. (b) Figure 2 illustrates that Tideos-Howto100m and Tideos-Ego4D focus primarily on human-centric activities, while Tideos-WebVid and Tideos-WordNet cover more scenarios. (c) Table 3 compares vocabulary sizes, revealing that Tideos-WordNet encompasses more objects.

---

### Decision · Program_Chairs · 2024-09-25

**Decision:**

Accept (spotlight)

**Comment:**

The paper initially received mixed scores, but during the AC-reviewers discussion phase, the reviewers reached a consensus that the work holds significant value due to its novelty, effectiveness, and potential impact. However, it is important to note that all three reviewers agree that incorporating the missing works identified by Reviewer kuEy would enhance the paper. Therefore, the AC also strongly recommends that the authors include these references in the camera-ready version.